# Variational Bayesian Monte Carlo
# with Noisy Likelihoods

**Luigi Acerbi**[*]
Department of Computer Science
University of Helsinki
`luigi.acerbi@helsinki.fi`

## Abstract

Variational Bayesian Monte Carlo (VBMC) is a recently introduced framework that uses Gaussian process surrogates to perform approximate Bayesian inference in models with *black-box*, non-cheap likelihoods. In this work, we extend VBMC to deal with noisy log-likelihood evaluations, such as those arising from simulation-based models. We introduce new 'global' acquisition functions, such as *expected information gain* (EIG) and *variational interquantile range* (VIQR), which are robust to noise and can be efficiently evaluated within the VBMC setting. In a novel, challenging, noisy-inference benchmark comprising of a variety of models with real datasets from computational and cognitive neuroscience, VBMC +VIQR achieves state-of-the-art performance in recovering the ground-truth posteriors and model evidence. In particular, our method vastly outperforms 'local' acquisition functions and other surrogate-based inference methods while keeping a small algorithmic cost. Our benchmark corroborates VBMC as a general-purpose technique for sample-efficient black-box Bayesian inference also with noisy models.

## 1 Introduction

Bayesian inference provides a principled framework for uncertainty quantification and model selection via computation of the posterior distribution over model parameters and of the model evidence [1, 2]. However, for many *black-box* models of interest in fields such as computational biology and neuroscience, (log-)likelihood evaluations are computationally expensive (thus limited in number) and noisy due to, e.g., simulation-based approximations [3, 4]. These features make standard techniques for approximate Bayesian inference such as Markov Chain Monte Carlo (MCMC) ineffective.

Variational Bayesian Monte Carlo (VBMC) is a recently proposed framework for Bayesian inference with non-cheap models [5, 6]. VBMC performs variational inference using a Gaussian process (GP [7]) as a statistical surrogate model for the expensive log posterior distribution. The GP model is refined via active sampling, guided by a 'smart' *acquisition function* that exploits uncertainty and other features of the surrogate. VBMC is particularly efficient thanks to a representation that affords fast integration via Bayesian quadrature [8, 9], and unlike other surrogate-based techniques it performs both posterior and model inference [5]. However, the original formulation of VBMC does not support noisy model evaluations, and recent work has shown that surrogate-based approaches that work well in the noiseless case may fail in the presence of even small amounts of noise [10].

In this work, we extend VBMC to deal robustly and effectively with noisy log-likelihood evaluations, broadening the class of models that can be estimated via the method. With our novel contributions, VBMC outperforms other state-of-the-art surrogate-based techniques for black-box Bayesian inference in the presence of noisy evaluations – in terms of speed, robustness and quality of solutions.

---

[*]Previous affiliation: Department of Basic Neuroscience, University of Geneva.

**Contributions** We make the following contributions: (1) we introduce several new acquisition functions for VBMC that explicitly account for noisy log-likelihood evaluations, and leverage the variational representation to achieve much faster evaluation than competing methods; (2) we introduce *variational whitening*, a technique to deal with non-axis aligned posteriors, which are otherwise potentially problematic for VBMC (and GP surrogates more in general) in the presence of noise; (3) we build a novel and challenging noisy-inference benchmark that includes five different models from computational and cognitive neuroscience, ranging from 3 to 9 parameters, and applied to real datasets, in which we test VBMC and other state-of-the-art surrogate-based inference techniques. The new features have been implemented in VBMC: https://github.com/lacerbi/vbmc.

**Related work** Our paper extends the VBMC framework [5, 6] by building on recent information-theoretical approaches to adaptive Bayesian quadrature [11], and on recent theoretical and empirical results for GP-surrogate Bayesian inference for simulation-based models [10, 12, 13]. For noiseless evaluations, previous work has used GP surrogates for estimation of posterior distributions [14–16] and Bayesian quadrature for calculation of the model evidence [9, 17–20]. Our method is also closely related to (noisy) Bayesian optimization [21–27]. A completely different approach, but worth mentioning for the similar goal, trains deep networks on simulated data to reconstruct approximate Bayesian posteriors from data or summary statistics thereof [28–31].

## 2 Variational Bayesian Monte Carlo (VBMC)

We summarize here the Variational Bayesian Monte Carlo (VBMC) framework [5]. If needed, we refer the reader to the Supplement for a recap of key concepts in variational inference, GPs and Bayesian quadrature. Let $f = \log p(\mathcal{D}|\boldsymbol{\theta})p(\boldsymbol{\theta})$ be the *target* log joint probability (unnormalized posterior), where $p(\mathcal{D}|\boldsymbol{\theta})$ is the model likelihood for dataset $\mathcal{D}$ and parameter vector $\boldsymbol{\theta} \in \mathcal{X} \subseteq \mathbb{R}^D$, and $p(\boldsymbol{\theta})$ the prior. We assume that only a limited number of log-likelihood evaluations are available, up to several hundreds. VBMC works by iteratively improving a variational approximation $q_{\boldsymbol{\phi}}(\boldsymbol{\theta})$, indexed by $\boldsymbol{\phi}$, of the true posterior density. In each iteration $t$, the algorithm:

1. Actively samples sequentially $n_{\text{active}}$ 'promising' new points, by iteratively maximizing a given acquisition function $a(\boldsymbol{\theta}) : \mathcal{X} \rightarrow \mathbb{R}$; for each selected point $\boldsymbol{\theta}_\star$ evaluates the target $\boldsymbol{y}_\star \equiv f(\boldsymbol{\theta}_\star)$ ($n_{\text{active}} = 5$ by default).
2. Trains a GP surrogate model of the log joint $f$, given the training set $\boldsymbol{\Xi}_t = \{\boldsymbol{\Theta}_t, \boldsymbol{y}_t\}$ of input points and their associated observed values so far.
3. Updates the variational posterior parameters $\boldsymbol{\phi}_t$ by optimizing the surrogate ELBO (variational lower bound on the model evidence) calculated via Bayesian quadrature.

This loop repeats until reaching a termination criterion (e.g., budget of function evaluations or lack of improvement over several iterations), and the algorithm returns both the variational posterior and posterior mean and variance of the ELBO. VBMC includes an initial *warm-up* stage to converge faster to regions of high posterior probability, before starting to refine the variational solution (see [5]).

### 2.1 Basic features

We briefly describe here basic features of the original VBMC framework [5] (see also Supplement).

**Variational posterior** The variational posterior is a flexible mixture of $K$ multivariate Gaussians, $q(\boldsymbol{\theta}) \equiv q_{\boldsymbol{\phi}}(\boldsymbol{\theta}) = \sum_{k=1}^K w_k \mathcal{N}\left(\boldsymbol{\theta}; \boldsymbol{\mu}_k, \sigma_k^2 \boldsymbol{\Sigma}\right)$, where $w_k$, $\boldsymbol{\mu}_k$, and $\sigma_k$ are, respectively, the mixture weight, mean, and scale of the $k$-th component; and $\boldsymbol{\Sigma}$ is a common diagonal covariance matrix $\boldsymbol{\Sigma} \equiv \text{diag}[\lambda^{(1)2}, \ldots, \lambda^{(D)2}]$. For a given $K$, the variational parameter vector is $\boldsymbol{\phi} \equiv (w_1, \ldots, w_K, \boldsymbol{\mu}_1, \ldots, \boldsymbol{\mu}_K, \sigma_1, \ldots, \sigma_K, \boldsymbol{\lambda})$. $K$ is set adaptively; fixed to $K = 2$ during warm-up, and then increasing each iteration if it leads to an improvement of the ELBO.

**Gaussian process model** In VBMC, the log joint $f$ is approximated by a GP surrogate model with a squared exponential (rescaled Gaussian) kernel, a Gaussian likelihood, and a *negative quadratic* mean function which ensures finiteness of the variational objective [5, 6]. In the original formulation, observations are assumed to be exact (non-noisy), so the GP likelihood only included a small observation noise $\sigma_{\text{obs}}^2$ for numerical stability [32]. GP hyperparameters are estimated initially via MCMC sampling [33], when there is larger uncertainty about the GP model, and later via a maximum-a-posteriori (MAP) estimate using gradient-based optimization (see [5] for details).

**The Evidence Lower Bound (ELBO)**  Using the GP surrogate model $f$, and for a given variational posterior $q_\phi$, the posterior mean of the surrogate ELBO can be estimated as

$$\mathbb{E}_{f|\Xi}\left[\text{ELBO}(\phi)\right] = \mathbb{E}_{f|\Xi}\left[\mathbb{E}_\phi\left[f\right]\right] + \mathcal{H}[q_\phi], \tag{1}$$

where $\mathbb{E}_{f|\Xi}\left[\mathbb{E}_\phi\left[f\right]\right]$ is the posterior mean of the expected log joint under the GP model, and $\mathcal{H}[q_\phi]$ is the entropy of the variational posterior. In particular, the expected log joint $\mathcal{G}$ takes the form

$$\mathcal{G}\left[q_\phi|f\right] \equiv \mathbb{E}_\phi\left[f\right] = \int q_\phi(\boldsymbol{\theta})f(\boldsymbol{\theta})d\boldsymbol{\theta}. \tag{2}$$

Crucially, the choice of variational family and GP representation affords closed-form solutions for the posterior mean and variance of Eq. 2 (and of their gradients) by means of Bayesian quadrature [8, 9]. The entropy of $q_\phi$ and its gradient are estimated via simple Monte Carlo and the reparameterization trick [34, 35], such that Eq. 1 can be optimized via stochastic gradient ascent [36].

**Acquisition function**  During the active sampling stage, new points to evaluate are chosen sequentially by maximizing a given *acquisition function* $a(\boldsymbol{\theta}) : \mathcal{X} \to \mathbb{R}$ constructed to represent useful search heuristics [37]. The VBMC paper introduced *prospective uncertainty sampling* [5],

$$a_{\text{pro}}(\boldsymbol{\theta}) = s_\Xi^2(\boldsymbol{\theta})q_\phi(\boldsymbol{\theta}) \exp\left(\overline{f}_\Xi(\boldsymbol{\theta})\right), \tag{3}$$

where $\overline{f}_\Xi(\boldsymbol{\theta})$ and $s_\Xi^2(\boldsymbol{\theta})$ are, respectively, the GP posterior latent mean and variance at $\boldsymbol{\theta}$ given the current training set $\Xi$. Effectively, $a_{\text{pro}}$ promotes selection of new points from regions of high probability density, as represented by the variational posterior and (exponentiated) posterior mean of the surrogate log-joint, for which we are also highly uncertain (high variance of the GP surrogate).

**Inference space**  The variational posterior and GP surrogate in VBMC are defined in an unbounded *inference space* equal to $\mathbb{R}^D$. Parameters that are subject to bound constraints are mapped to the inference space via a shifted and rescaled logit transform, with an appropriate Jacobian correction to the log-joint. Solutions are transformed back to the original space via a matched inverse transform, e.g., a shifted and rescaled logistic function for bound parameters (see [5, 38]).

## 2.2  Variational whitening

One issue of the standard VBMC representation of both the variational posterior and GP surrogate is that it is axis-aligned, which makes it ill-suited to deal with highly correlated posteriors. As a simple and inexpensive solution, we introduce here *variational whitening*, which consists of a linear transformation $\mathbf{W}$ of the inference space (a rotation and rescaling) such that the variational posterior $q_\phi$ obtains unit diagonal covariance matrix. Since $q_\phi$ is a mixture of Gaussians in inference space, its covariance matrix $\mathbf{C}_\phi$ is available in closed form and we can calculate the whitening transform $\mathbf{W}$ by performing a singular value decomposition (SVD) of $\mathbf{C}_\phi$. We start performing variational whitening a few iterations after the end of warm-up, and then at increasingly more distant intervals. By default we use variational whitening with all variants of VBMC tested in this paper; see the Supplement for an ablation study demonstrating its usefulness and for further implementation details.

## 3  VBMC with noisy likelihood evaluations

Extending the framework described in Section 2, we now assume that evaluations of the log-likelihood $y_n$ can be noisy, that is

$$y_n = f(\boldsymbol{\theta}_n) + \sigma_{\text{obs}}(\boldsymbol{\theta}_n)\varepsilon_n, \qquad \varepsilon_n \overset{\text{i.i.d.}}{\sim} \mathcal{N}(0, 1), \tag{4}$$

where $\sigma_{\text{obs}} : \mathcal{X} \to [\sigma_{\text{min}}, \infty)$ is a function of the input space that determines the standard deviation (SD) of the observation noise. For this work, we use $\sigma_{\text{min}}^2 = 10^{-5}$ and we assume that the evaluation of the log-likelihood at $\boldsymbol{\theta}_n$ returns both $y_n$ and a reasonable estimate $(\widehat{\sigma}_{\text{obs}})_n$ of $\sigma_{\text{obs}}(\boldsymbol{\theta}_n)$. Here we estimate $\sigma_{\text{obs}}(\boldsymbol{\theta})$ outside the training set via a nearest-neighbor approximation (see Supplement), but more sophisticated methods could be used (e.g., by training a GP model on $\sigma_{\text{obs}}(\boldsymbol{\theta}_n)$ [39]).

The *synthetic likelihood* (SL) technique [3, 4] and *inverse binomial sampling* (IBS) [40, 41] are examples of log-likelihood estimation methods for simulation-based models that satisfy the assumptions of our observation model (Eq. 4). Recent work demonstrated empirically that log-SL estimates are approximately normally distributed, and their SD can be estimated accurately via bootstrap [10].

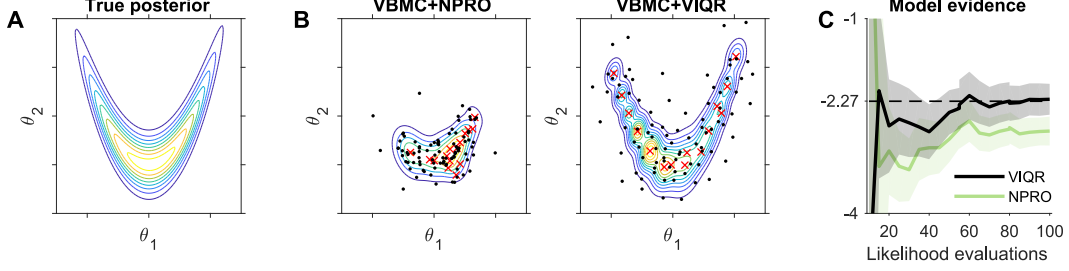

Figure 1: VBMC **with noisy likelihoods. A.** True target pdf ($D = 2$). We assume noisy log-likelihood evaluations with $\sigma_{\mathrm{obs}} = 1$. **B.** Contour plots of the variational posterior after 100 likelihood evaluations, with the noise-adjusted $a_{\mathrm{npro}}$ acquisition function (left) and the newly proposed $a_{\mathrm{VIQR}}$ (right). Red crosses indicate the centers of the variational mixture components, black dots are the training samples. **C.** ELBO as a function of likelihood evaluations. Shaded area is 95% CI of the ELBO estimated via Bayesian quadrature. Dashed line is the true log marginal likelihood (LML).

IBS is a recently reintroduced statistical technique that produces both normally-distributed, unbiased estimates of the log-likelihood and calibrated estimates of their variance [41].

In the rest of this section, we describe several new acquisition functions for VBMC specifically designed to deal with noisy log-likelihood evaluations. Figure 1 shows VBMC at work in a toy noisy scenario (a 'banana' 2D posterior), for two acquisition functions introduced in this section.

**Predictions with noisy evaluations**    A useful quantity for this section is $s^2_{\Xi \cup \boldsymbol{\theta}_\star}(\boldsymbol{\theta})$, the predicted posterior GP variance at $\boldsymbol{\theta}$ if we make a function evaluation at $\boldsymbol{\theta}_\star$, with $y_\star$ distributed according to the posterior *predictive* distribution (that is, inclusive of observation noise $\sigma_{\mathrm{obs}}(\boldsymbol{\theta}_\star)$), given training data $\Xi$. Conveniently, $s^2_{\Xi \cup \boldsymbol{\theta}_\star}(\boldsymbol{\theta})$ can be expressed in closed form as

$$s^2_{\Xi \cup \boldsymbol{\theta}_\star}(\boldsymbol{\theta}) = s^2_{\Xi}(\boldsymbol{\theta}) - \frac{C^2_{\Xi}(\boldsymbol{\theta}, \boldsymbol{\theta}_\star)}{C_{\Xi}(\boldsymbol{\theta}_\star, \boldsymbol{\theta}_\star) + \sigma^2_{\mathrm{obs}}(\boldsymbol{\theta}_\star)}, \tag{5}$$

where $C_{\Xi}(\cdot, \cdot)$ denotes the GP posterior covariance (see [10, Lemma 5.1], and also [13, 42]).

### 3.1   Noisy prospective uncertainty sampling

The rationale behind $a_{\mathrm{pro}}$ (Eq. 3) and similar heuristic 'uncertainty sampling' acquisition functions [6, 18] is to evaluate the log joint where the pointwise variance of the integrand in the expected log joint (as per Eq. 2, or variants thereof) is maximum. For noiseless evaluations, this choice is equivalent to maximizing the *variance reduction* of the integrand after an observation. Considering the GP posterior variance reduction, $\Delta s^2_{\Xi}(\boldsymbol{\theta}) \equiv s^2_{\Xi}(\boldsymbol{\theta}) - s^2_{\Xi \cup \boldsymbol{\theta}}(\boldsymbol{\theta})$, we see that, in the absence of observation noise, $s^2_{\Xi \cup \boldsymbol{\theta}}(\boldsymbol{\theta}) = 0$ and $\Delta s^2(\boldsymbol{\theta})_{\Xi} = s^2_{\Xi}(\boldsymbol{\theta})$. Thus, a natural generalization of uncertainty sampling to the noisy case is obtained by switching the GP posterior variance in Eq. 3 to the GP posterior variance *reduction*. Improving over the original uncertainty sampling, this generalization accounts for potential observation noise at the candidate location.

Following this reasoning, we generalize uncertainty sampling to noisy observations by defining the *noise-adjusted prospective uncertainty sampling* acquisition function,

$$a_{\mathrm{npro}}(\boldsymbol{\theta}) = \Delta s^2_{\Xi}(\boldsymbol{\theta}) q_\phi(\boldsymbol{\theta}) \exp\left(\overline{f}_{\Xi}(\boldsymbol{\theta})\right) = \left(\frac{s^2_{\Xi}(\boldsymbol{\theta})}{s^2_{\Xi}(\boldsymbol{\theta}) + \sigma^2_{\mathrm{obs}}(\boldsymbol{\theta})}\right) s^2_{\Xi}(\boldsymbol{\theta}) q_\phi(\boldsymbol{\theta}) \exp\left(\overline{f}_{\Xi}(\boldsymbol{\theta})\right), \tag{6}$$

where we used Eq. 5 to calculate $s^2_{\Xi \cup \boldsymbol{\theta}}(\boldsymbol{\theta})$. Comparing Eq. 6 to Eq. 3, we see that $a_{\mathrm{npro}}$ has an additional multiplicative term that accounts for the residual variance due to a potentially noisy observation. As expected, it is easy to see that $a_{\mathrm{npro}}(\boldsymbol{\theta}) \to a_{\mathrm{pro}}(\boldsymbol{\theta})$ for $\sigma_{\mathrm{obs}}(\boldsymbol{\theta}) \to 0$.

While $a_{\mathrm{npro}}$ and other forms of uncertainty sampling operate pointwise on the posterior *density*, we consider next *global* (integrated) acquisition functions that account for non-local changes in the GP surrogate model when making a new observation, thus driven by uncertainty in posterior *mass*.

### 3.2   Expected information gain (EIG)

A principled information-theoretical approach suggests to sample points that maximize the *expected information gain* (EIG) about the integral of interest (Eq. 2). Following recent work on multi-source

active-sampling Bayesian quadrature [11], we can do so by choosing the next location $\boldsymbol{\theta}_\star$ that maximizes the *mutual information* $I\left[\mathcal{G}; y_\star\right]$ between the expected log joint $\mathcal{G}$ and a new (unknown) observation $y_\star$. Since all involved quantities are jointly Gaussian, we obtain

$$a_{\text{EIG}}(\boldsymbol{\theta}) = -\frac{1}{2}\log\left(1 - \rho^2(\boldsymbol{\theta})\right), \qquad \text{with } \rho(\boldsymbol{\theta}) \equiv \frac{\mathbb{E}_\phi\left[C_{\boldsymbol{\Xi}}(f(\cdot), f(\boldsymbol{\theta}))\right]}{\sqrt{v_{\boldsymbol{\Xi}}(\boldsymbol{\theta})\mathbb{V}_{f|\boldsymbol{\Xi}}[\mathcal{G}]}}, \qquad (7)$$

where $\rho(\cdot)$ is the *scalar correlation* [11], $v_{\boldsymbol{\Xi}}(\cdot)$ the GP posterior predictive variance (including observation noise), and $\mathbb{V}_{f|\boldsymbol{\Xi}}[\mathcal{G}]$ the posterior variance of the expected log joint – all given the current training set $\boldsymbol{\Xi}$. The scalar correlation in Eq. 7 has a closed-form solution thanks to Bayesian quadrature (see Supplement for derivations).

### 3.3  Integrated median / variational interquantile range (IMIQR/ VIQR)

Järvenpää and colleagues [10] recently proposed the *interquantile range* (IQR) as a robust estimate of the uncertainty of the unnormalized posterior, as opposed to the variance, and derived the *integrated median interquantile range* (IMIQR) acquisition function from Bayesian decision theory,

$$a_{\text{IMIQR}}(\boldsymbol{\theta}) = -2\int_{\mathcal{X}}\exp\left(\overline{f}_{\boldsymbol{\Xi}}(\boldsymbol{\theta}')\right)\sinh\left(us_{\boldsymbol{\Xi}\cup\boldsymbol{\theta}}(\boldsymbol{\theta}')\right)d\boldsymbol{\theta}', \qquad (8)$$

where $u \equiv \Phi^{-1}(p_u)$, with $\Phi$ the standard normal CDF and $p_u \in (0.5, 1)$ a chosen quantile (we use $p_u = 0.75$ as in [10]); $\sinh(z) = (\exp(z) - \exp(-z))/2$ for $z \in \mathbb{R}$ is the hyperbolic sine; and $s_{\boldsymbol{\Xi}\cup\boldsymbol{\theta}}(\boldsymbol{\theta}')$ denotes the predicted posterior standard deviation after observing the function at $\boldsymbol{\theta}'$, as per Eq. 5. However, the integral in Eq. 8 is intractable, and thus needs to be approximated at a significant computational cost (e.g., via MCMC and importance sampling [10]).

Instead, we note that the term $\exp\left(\overline{f}_{\boldsymbol{\Xi}}\right)$ in Eq. 8 represents the joint distribution as modeled via the GP surrogate, which VBMC further approximates with the variational posterior $q_\phi$ (up to a normalization constant). Thus, we exploit the variational approximation of VBMC to propose here the *variational* (integrated median) *interquantile range* (VIQR) acquisition function,

$$a_{\text{VIQR}}(\boldsymbol{\theta}) = -2\int_{\mathcal{X}} q_\phi(\boldsymbol{\theta}')\sinh\left(us_{\boldsymbol{\Xi}\cup\boldsymbol{\theta}}(\boldsymbol{\theta}')\right)d\boldsymbol{\theta}', \qquad (9)$$

where we replaced the surrogate posterior in Eq. 8 with its corresponding variational posterior. Crucially, Eq. 9 can be approximated very cheaply via simple Monte Carlo by drawing $N_{\text{viqr}}$ samples from $q_\phi$ (we use $N_{\text{viqr}} = 100$). In brief, $a_{\text{VIQR}}$ obtains a computational advantage over $a_{\text{IMIQR}}$ at the cost of adding a layer of approximation in the acquisition function ($q_\phi \approx \exp\left(\overline{f}_{\boldsymbol{\Xi}}\right)$), but it otherwise follows from the same principles. Whether this approximation is effective in practice is an empirical question that we address in the next section.

## 4  Experiments

We tested different versions of VBMC and other surrogate-based inference algorithms on a novel benchmark problem set consisting of a variety of computational models applied to real data (see Section 4.1). For each problem, the goal of inference is to approximate the posterior distribution and the log marginal likelihood (LML) with a fixed budget of likelihood evaluations.

**Algorithms**  In this work, we focus on comparing new acquisition functions for VBMC which support noisy likelihood evaluations, that is $a_{\text{npro}}$, $a_{\text{EIG}}$, $a_{\text{IMIQR}}$ and $a_{\text{VIQR}}$ as described in Section 3 (denoted as VBMC plus, respectively, NPRO, EIG, IMIQR or VIQR). As a strong baseline for posterior estimation, we test a state-of-the-art technique for Bayesian inference via GP surrogates, which also uses $a_{\text{IMIQR}}$ [10] (GP-IMIQR). GP-IMIQR was recently shown to decisively outperform several other surrogate-based methods for posterior estimation in the presence of noisy likelihoods [10]. For model evidence evaluation, to our knowledge no previous surrogate-based technique explicitly supports noisy evaluations. We test as a baseline *warped sequential active Bayesian integration* (WSABI [18]), a competitive method in a previous noiseless comparison [5], adapted here for our benchmark (see Supplement). For each algorithm, we use the same default settings across problems. We do not consider here non-surrogate based methods, such as Monte Carlo and importance sampling, which performed poorly with a limited budget of likelihood evaluations already in the noiseless case [5].

**Procedure** For each problem, we allow a budget of $50 \times (D+2)$ likelihood evaluations. For each algorithm, we performed 100 runs per problem with random starting points, and we evaluated performance with several metrics (see Section 4.2). For each metric, we report as a function of likelihood evaluations the median and 95% CI of the median calculated by bootstrap (see Supplement for a 'worse-case' analysis of performance). For algorithms other than VBMC, we only report metrics they were designed for (posterior estimation for GP-IMIQR, model evidence for WSABI).

**Noisy log-likelihoods** For a given data set, model and parameter vector $\boldsymbol{\theta}$, we obtain noisy evaluations of the log-likelihood through different methods, depending on the problem. In the *synthetic likelihood* (SL) approach, we run $N_{\text{sim}}$ simulations for each evaluation, and estimate the log-likelihood of summary statistics of the data under a multivariate normal assumption [3, 4, 10]. With *inverse binomial sampling* (IBS), we obtain unbiased estimates of the log-likelihood of an entire data set by sampling from the model until we obtain a 'hit' for each data point [40, 41]; we repeat the process $N_{\text{rep}}$ times and average the estimates for higher precision. Finally, for a few analyses we 'emulate' noisy evaluations by adding i.i.d. Gaussian noise to deterministic log-likelihoods. Despite its simplicity, the 'emulated noise' approach is statistically similar to IBS, as IBS estimates are unbiased, normally-distributed, and with near-constant variance across the parameter space [41].

## 4.1 Benchmark problems

The benchmark problem set consists of a common test simulation model (the Ricker model [3]) and five models with real data from various branches of computational and cognitive neuroscience. Some models are applied to multiple datasets, for a total of nine inference problems with $3 \leq D \leq 9$ parameters. Each problem provides a target noisy log-likelihood, and for simplicity we assume a uniform prior over a bounded interval for each parameter. For the purpose of this benchmark, we chose tractable models so that we could compute ground-truth posteriors and model evidence via extensive MCMC sampling. We now briefly describe each model; see Supplement for more details.

**Ricker** The Ricker model is a classic population model used in computational ecology [3]. The population size $N_t$ evolves according to a discrete-time stochastic process $N_{t+1} = rN_t \exp(-N_t + \varepsilon_t)$, for $t = 1, \ldots, T$, with $\varepsilon_t \overset{\text{i.i.d.}}{\sim} \mathcal{N}(0, \sigma_\varepsilon^2)$ and $N_0 = 1$. At each time point, we have access to a noisy measurement $z_t$ of the population size $N_t$ with Poisson observation model $z_t \sim \text{Poisson}(\phi N_t)$. The model parameters are $\boldsymbol{\theta} = (\log(r), \phi, \sigma_\varepsilon)$. We generate a dataset of observations $\boldsymbol{z} = (z_t)_{t=1}^T$ using the "true" parameter vector $\boldsymbol{\theta}_{\text{true}} = (3.8, 10, 0.3)$ with $T = 50$, as in [10]. We estimate the log-likelihood via the log-SL approach using the same 13 summary statistics as in [3, 4, 10, 25], with $N_{\text{sim}} = 100$ simulations per evaluation, which yields $\sigma_{\text{obs}}(\boldsymbol{\theta}_{\text{MAP}}) \approx 1.3$, where $\boldsymbol{\theta}_{\text{MAP}}$ is the maximum-a-posteriori (MAP) parameter estimate found via optimization.

**Attentional drift-diffusion model (aDDM)** The *attentional drift-diffusion model* (aDDM) is a seminal model for value-based decision making between two items with ratings $r_{\text{A}}$ and $r_{\text{B}}$ [43]. At each time step $t$, the decision variable $z_t$ is assumed to follow a stochastic diffusion process

$$z_0 = 0, \qquad z_{t+\delta t} = z_t + d\left(\beta^{a_t}r_{\text{A}} - \beta^{(1-a_t)}r_{\text{B}}\right)\delta t + \varepsilon_t, \qquad \varepsilon_t \overset{\text{i.i.d.}}{\sim} \mathcal{N}\left(0, \sigma_\varepsilon^2 \delta t\right), \qquad (10)$$

where $\varepsilon_t$ is the diffusion noise; $d$ is the drift rate; $\beta \in [0, 1]$ is the attentional bias factor; and $a_t = 1$ (resp., $a_t = 0$) if the subject is fixating item A (resp., item B) at time $t$. Diffusion continues until the decision variable hits the boundary $|z_t| \geq 1$, which induces a choice (A for +1, B for -1). We include a lapse probability $\lambda$ of a random choice at a uniformly random time over the maximum trial duration, and set $\delta t = 0.1$ s. The model has parameters $\boldsymbol{\theta} = (d, \beta, \sigma_\varepsilon, \lambda)$. We fit choices and reaction times of two subjects (S1 and S2) from [43] using IBS with $N_{\text{rep}} = 500$, which produces $\sigma_{\text{obs}}(\boldsymbol{\theta}_{\text{MAP}}) \approx 2.8$.

**Bayesian timing** We consider a popular model of Bayesian time perception [44, 45]. In each trial of a sensorimotor timing task, human subjects had to reproduce the time interval $\tau$ between a click and a flash, with $\tau \sim \text{Uniform}[0.6, 0.975]$ s [45]. We assume subjects had only access to a noisy sensory measurement $t_{\text{s}} \sim \mathcal{N}\left(\tau, w_{\text{s}}^2 \tau^2\right)$, and their reproduced time $t_{\text{m}}$ was affected by motor noise, $t_{\text{m}} \sim \mathcal{N}\left(\tau_\star, w_m^2 \tau_\star^2\right)$, where $w_{\text{s}}$ and $w_{\text{m}}$ are Weber's fractions. We assume subjects estimated $\tau_\star$ by combining their sensory likelihood with an approximate Gaussian prior over time intervals, $\mathcal{N}\left(\tau; \mu_{\text{p}}, \sigma_{\text{p}}^2\right)$, and took the mean of the resulting Bayesian posterior. For each trial we also consider a probability $\lambda$ of a 'lapse' (e.g., a misclick) producing a response $t_{\text{m}} \sim \text{Uniform}[0, 2]$ s. Model parameters are $\boldsymbol{\theta} = (w_{\text{s}}, w_{\text{m}}, \mu_{\text{p}}, \sigma_{\text{p}}, \lambda)$. We fit timing responses (discretized with $\delta t_{\text{m}} = 0.02$ s) of a representative subject from [45] using IBS with $N_{\text{rep}} = 500$, which yields $\sigma_{\text{obs}}(\boldsymbol{\theta}_{\text{MAP}}) \approx 2.2$.

**Multisensory causal inference (CI)**   *Causal inference* (CI) in multisensory perception denotes the problem the brain faces when deciding whether distinct sensory cues come from the same source [46]. We model a visuo-vestibular CI experiment in which human subjects, sitting in a moving chair, were asked in each trial whether the direction of movement $s_{\text{vest}}$ matched the direction $s_{\text{vis}}$ of a looming visual field [47]. We assume subjects only have access to noisy sensory measurements $z_{\text{vest}} \sim \mathcal{N}\left(s_{\text{vest}}, \sigma_{\text{vest}}^2\right)$, $z_{\text{vis}} \sim \mathcal{N}\left(s_{\text{vis}}, \sigma_{\text{vis}}^2(c)\right)$, where $\sigma_{\text{vest}}$ is the vestibular noise and $\sigma_{\text{vis}}(c)$ is the visual noise, with $c \in \{c_{\text{low}}, c_{\text{med}}, c_{\text{high}}\}$ distinct levels of visual coherence adopted in the experiment. We model subjects' responses with a heuristic 'Fixed' rule that judges the source to be the same if $|z_{\text{vis}} - z_{\text{vest}}| < \kappa$, plus a probability $\lambda$ of giving a random response (*lapse*) [47]. Model parameters are $\boldsymbol{\theta} = (\sigma_{\text{vest}}, \sigma_{\text{vis}}(c_{\text{low}}), \sigma_{\text{vis}}(c_{\text{med}}), \sigma_{\text{vis}}(c_{\text{high}}), \kappa, \lambda)$. We fit datasets from two subjects (S1 and S2) from [47] using IBS with $N_{\text{rep}} = 200$ repeats, which yields $\sigma_{\text{obs}}(\boldsymbol{\theta}_{\text{MAP}}) \approx 1.3$ for both datasets.

**Neuronal selectivity**   We consider a computational model of neuronal orientation selectivity in visual cortex [48] used in previous optimization and inference benchmarks [5, 6, 26]. It is a linear-nonlinear-linear-nonlinear (LN-LN) cascade model which combines effects of filtering, suppression, and response nonlinearity whose output drives the firing rate of an inhomogeneous Poisson process (details in [48]). The restricted model has $D = 7$ free parameters which determine features such as the neuron's preferred direction of motion and spatial frequency. We fit the neural recordings of one V1 and one V2 cell from [48]. For the purpose of this 'noisy' benchmark, we compute the log-likelihood exactly and add i.i.d. Gaussian noise to each log-likelihood evaluation with $\sigma_{\text{obs}}(\boldsymbol{\theta}) = 2$.

**Rodent 2AFC**   We consider a sensory-history-dependent model of rodent decision making in a two-alternative forced choice (2AFC) task. In each trial, rats had to discriminate the amplitudes $s_{\text{L}}$ and $s_{\text{R}}$ of auditory tones presented, respectively, left and right [49, 50]. The rodent's choice probability is modeled as $P(\text{Left}) = \lambda/2 + (1 - \lambda)/(1 + e^{-A})$ where $\lambda$ is a lapse probability and

$$A = w_0 + w_{\text{c}} b_{\text{c}}^{(-1)} + w_{\overline{s}} \overline{s} + \sum_{t=0}^{2} \left( w_{\text{L}}^{(-t)} s_{\text{L}}^{(-t)} + w_{\text{R}}^{(-t)} s_{\text{R}}^{(-t)} \right), \tag{11}$$

where $w_{\text{L}}^{(-t)}$ and $w_{\text{R}}^{(-t)}$ are coefficients of the $s_{\text{L}}$ and $s_{\text{R}}$ regressors, respectively, from $t$ trials back; $b_{\text{c}}^{(-1)}$ is the correct side on the previous trial (L $= +1$, R $= -1$), used to capture the win-stay/lose-switch strategy; $\overline{s}$ is a long-term history regressor (an exponentially-weighted running mean of past stimuli with time constant $\tau$); and $w_0$ is the bias. This choice of regressors best described rodents' behavior in the task [49]. We fix $\lambda = 0.02$ and $\tau = 20$ trials, thus leaving $D = 9$ free parameters $\boldsymbol{\theta} = (w_0, w_{\text{c}}, w_{\overline{s}}, \boldsymbol{w}_{\text{L}}^{(0,-1,-2)}, \boldsymbol{w}_{\text{R}}^{(0,-1,-2)})$. We fit $10^4$ trials from a representative subject dataset [50] using IBS with $N_{\text{rep}} = 500$, which produces $\sigma_{\text{obs}}(\boldsymbol{\theta}_{\text{MAP}}) \approx 3.18$.

## 4.2   Results

To assess the model evidence approximation, Fig. 2 shows the absolute difference between true and estimated log marginal likelihood ('LML loss'), using the ELBO as a proxy for VBMC. Differences in LML of 10+ points are often considered 'decisive evidence' in a model comparison [51], while differences $\ll 1$ are negligible; so for practical usability of a method we aim for a LML loss $< 1$.

As a measure of loss to judge the quality of the posterior approximation, Fig. 3 shows the mean marginal total variation distance (MMTV) between approximate posterior and ground truth. Given two pdfs $p$ and $q$, we define $\text{MMTV}(p, q) = \frac{1}{2D} \sum_{i=1}^{D} \int |p_i(x_i) - q_i(x_i)| \, dx_i$, where $p_i$ and $q_i$ denote the marginal densities along the $i$-th dimension. Since the MMTV only looks at differences in the marginals, we also examined the "Gaussianized" symmetrized Kullback-Leibler divergence (gsKL), a metric sensitive to differences in mean and covariance [5]. We found that MMTV and gsKL follow qualitatively similar trends, so we show the latter in the Supplement.

First, our results confirm that, in the presence of noisy log-likelihoods, methods that use 'global' acquisition functions largely outperform methods that use pointwise estimates of uncertainty, as noted in [10]. In particular, 'uncertainty sampling' acquisition functions are unusable with VBMC in the presence of noise, exemplified here by the poor performance of VBMC-NPRO (see also Supplement for further tests). WSABI shows the worst performance here due to a GP representation (the square root transform) which interacts badly with noise on the log-likelihood. Previous state-of-the art method GP-IMIQR performs well with a simple synthetic problem (Ricker), but fails on complex scenarios such as Rodent 2AFC, Neuronal selectivity, or Bayesian timing, likely due to excessive exploration

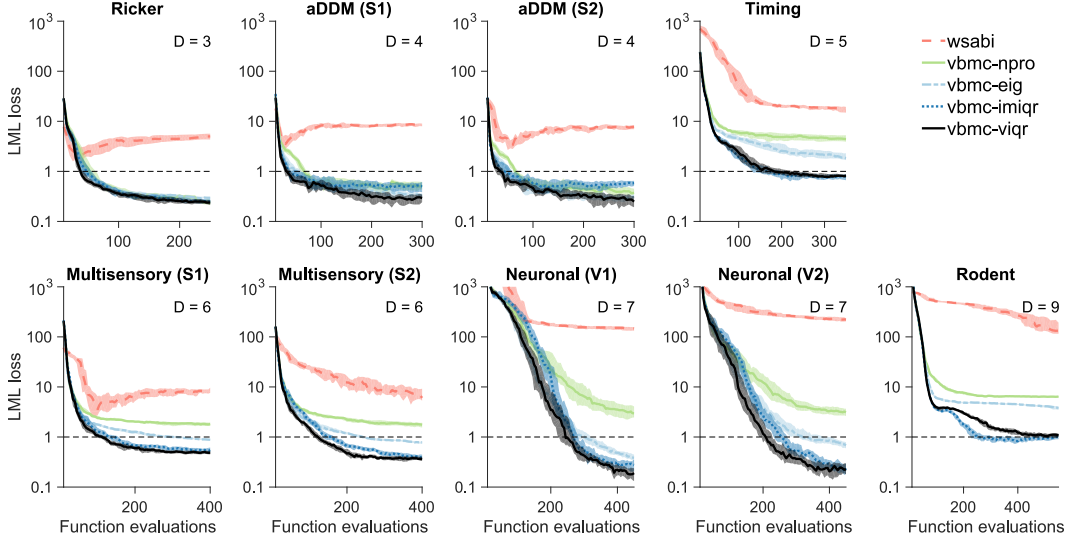

Figure 2: **Model evidence loss.** Median absolute error of the log marginal likelihood (LML) estimate with respect to ground truth, as a function of number of likelihood evaluations, on different problems. A desirable error is below 1 (dashed line). Shaded areas are 95% CI of the median across 100 runs.

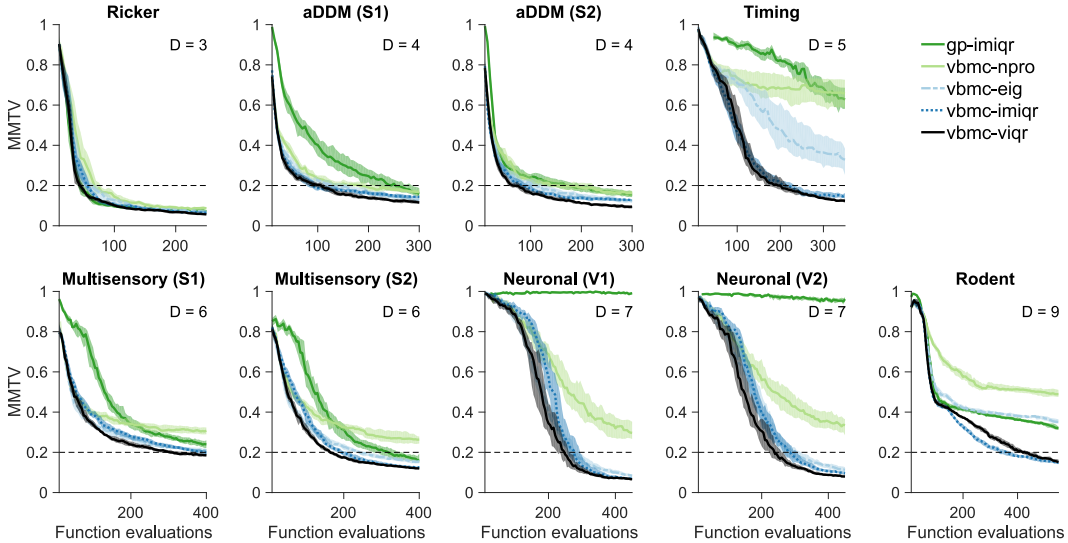

Figure 3: **Posterior estimation loss (MMTV).** Median mean marginal total variation distance (MMTV) between the algorithm's posterior and ground truth, as a function of number of likelihood evaluations. A desirable target (dashed line) is less than 0.2, corresponding to more than 80% overlap between true and approximate posterior marginals (on average across model parameters).

(see Supplement). VBMC-EIG performs reasonably well on most problems, but also struggles on Rodent 2AFC and Bayesian timing. Overall, VBMC-IMIQR and VBMC-VIQR systematically show the best and most robust performance, with VBMC-VIQR marginally better on most problems, except Rodent 2AFC. Both achieve good approximations of the model evidence and of the true posteriors within the limited budget (see Supplement for comparisons with ground-truth posteriors).

Table 1 compares the average algorithmic overhead of methods based on $a_{\text{IMIQR}}$ and $a_{\text{VIQR}}$, showing the computational advantage of the variational approach of VBMC-VIQR.

Then, we looked at how robust different methods are to different degrees of log-likelihood noise. We considered three benchmark problems for which we could easily compute the log-likelihood exactly. For each problem, we emulated different levels of noise by adding Gaussian observation noise to

Table 1: Average algorithmic overhead per likelihood evaluation (in seconds) over a full run, assessed on a single-core reference machine (mean $\pm$ 1 SD across 100 runs).

| Algorithm | Model | | | | | |
| | Ricker | aDDM | Timing | Multisensory | Neuronal | Rodent |
| --- | --- | --- | --- | --- | --- | --- |
| VBMC-VIQR | **1.5 ± 0.1** | **1.5 ± 0.1** | **1.8 ± 0.2** | **2.0 ± 0.2** | **2.8 ± 0.8** | **2.6 ± 0.2** |
| VBMC-IMIQR | 5.5 ± 0.5 | 5.1 ± 0.3 | 5.8 ± 0.6 | 5.6 ± 0.3 | 6.5 ± 1.3 | 5.6 ± 0.4 |
| GP-IMIQR | 15.6 ± 0.9 | 16.0 ± 1.7 | 17.1 ± 1.2 | 26.3 ± 1.8 | 29.6 ± 2.8 | 40.1 ± 2.1 |

exact log-likelihood evaluations, with $\sigma_{\text{obs}} \in [0, 7]$ (see Fig. 4). Most algorithms only perform well with no or very little noise, whereas the performance of VBMC-VIQR (and, similarly, VBMC-IMIQR) degrades gradually with increasing noise. For these two algorithms, acceptable results can be reached for $\sigma_{\text{obs}}$ as high as $\approx 7$, although for best results even with hard problems we would recommend $\sigma_{\text{obs}} \lesssim 3$. We see that the Neuronal problem is particularly hard, with both WSABI and GP-IMIQR failing to converge altogether even in the absence of noise.

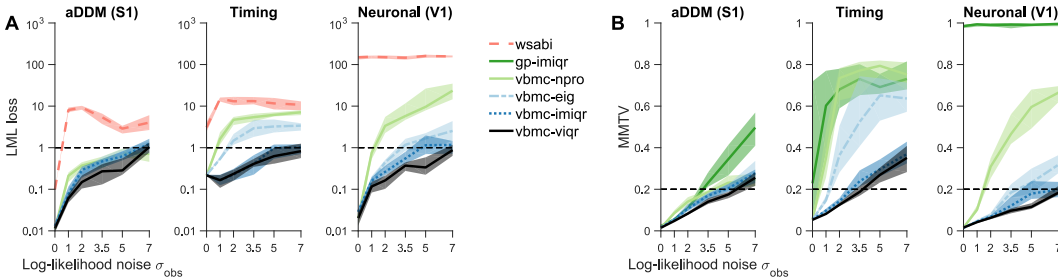

Figure 4: **Noise sensitivity.** Final performance metrics of all algorithms with respect to ground truth, as a function of log-likelihood observation noise $\sigma_{\text{obs}}$, for different problems. For all metrics, we plot the median after $50 \times (D+2)$ log-likelihood evaluations, and shaded areas are 95 % CI of the median across 100 runs. **A.** Absolute error of the log marginal likelihood (LML) estimate. **B.** Mean marginal total variation distance (MMTV).

Lastly, we tested how robust VBMC-VIQR is to imprecise estimates of the observation noise, $\widehat{\sigma}_{\text{obs}}(\boldsymbol{\theta})$. We reran VBMC-VIQR on the three problems of Fig. 4 while drawing $\widehat{\sigma}_{\text{obs}} \sim \text{Lognormal}\left(\ln \sigma_{\text{obs}}, \sigma_\sigma^2\right)$ for increasing values of noise-of-estimating-noise, $\sigma_\sigma \geq 0$. We found that at worst the performance of VBMC degrades only by $\sim 25\%$ with $\sigma_\sigma$ up to $0.4$ (i.e., $\widehat{\sigma}_{\text{obs}}$ roughly between $0.5 - 2.2$ times the true value); showing that VBMC is robust to imprecise noise estimates (see Supplement for details).

## 5 Conclusions

In this paper, we addressed the problem of approximate Bayesian inference with only a limited budget of noisy log-likelihood evaluations. For this purpose, we extended the VBMC framework to work in the presence of noise by testing several new acquisition functions and by introducing variational whitening for a more accurate posterior approximation. We showed that with these new features VBMC achieves state-of-the-art inference performance on a novel challenging benchmark that uses a variety of models and real data sets from computational and cognitive neuroscience, covering areas such as neuronal modeling, human and rodent psychophysics, and value-based decision-making.

Our benchmark also revealed that common synthetic test problems, such as the Ricker and g-and-k models (see Supplement for the latter), may be too simple for surrogate-based methods, as good performance on these problems (e.g., GP-IMIQR) may not generalize to real models and datasets.

In conclusion, our extensive analyses show that VBMC with the $a_{\text{VIQR}}$ acquisition function is very effective for approximate Bayesian inference with noisy log-likelihoods, with up to $\sigma_{\text{obs}} \approx 3$, and models up to $D \lesssim 10$ and whose evaluation take about a few seconds or more. Future work should focus on improving the flexibility of the GP representation, scaling the method to higher dimensions, and investigating theoretical guarantees for the VBMC algorithm.

## Broader Impact

We believe this work has the potential to lead to net-positive improvements in the research community and more broadly in society at large. First, this paper makes Bayesian inference accessible to non-cheap models with noisy log-likelihoods, allowing more researchers to express uncertainty about their models and model parameters of interest in a principled way; with all the advantages of proper uncertainty quantification [2]. Second, with the energy consumption of computing facilities growing incessantly every hour, it is our duty towards the environment to look for ways to reduce the carbon footprint of our algorithms [52]. In particular, traditional methods for approximate Bayesian inference can be extremely sample-inefficient. The 'smart' sample-efficiency of VBMC can save a considerable amount of resources when model evaluations are computationally expensive.

Failures of VBMC can return largely incorrect posteriors and values of the model evidence, which if taken at face value could lead to wrong conclusions. This failure mode is not unique to VBMC, but a common problem of all approximate inference techniques (e.g., MCMC or variational inference [2,53]). VBMC returns uncertainty on its estimate and comes with a set of diagnostic functions which can help identify issues. Still, we recommend the user to follow standard good practices for validation of results, such as posterior predictive checks, or comparing results from different runs.

Finally, in terms of ethical aspects, our method – like any general, black-box inference technique – will reflect (or amplify) the explicit and implicit biases present in the models and in the data, especially with insufficient data [54]. Thus, we encourage researchers in potentially sensitive domains to explicitly think about ethical issues and consequences of the models and data they are using.

## Acknowledgments and Disclosure of Funding

We thank Ian Krajbich for sharing data for the aDDM model; Robbe Goris for sharing data and code for the neuronal model; Marko Järvenpää and Alexandra Gessner for useful discussions about their respective work; Nisheet Patel for helpful comments on an earlier version of this manuscript; and the anonymous reviewers for constructive remarks.

This work has utilized the NYU IT High Performance Computing resources and services. This work was partially supported by the Swiss National Foundation (grant number 31003A_165831) and by the University of Helsinki (Faculty of Science), through grants of the Academy of Finland. We also thank the Academy of Finland Flagship programme: Finnish Center for Artificial Intelligence (FCAI). The funders had no role in study design, data collection and analysis, decision to publish, or preparation of the manuscript.

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
