[Supplementary Material]

# Variational Bayesian Monte Carlo
# with Noisy Likelihoods —
# Supplementary Material

**Luigi Acerbi**[*]
Department of Computer Science
University of Helsinki
`luigi.acerbi@helsinki.fi`

In this Supplement we include a number of derivations, implementation details, and additional results omitted from the main text.

Code used to generate the results and figures in the paper is available at `https://github.com/lacerbi/infbench`. The VBMC algorithm with added support for noisy models is available at `https://github.com/lacerbi/vbmc`.

## Contents

---

[*]Previous affiliation: Department of Basic Neuroscience, University of Geneva.

# A   Background information

For ease of reference, in this Section we recap the three key theoretical ingredients used to build the Variational Bayesian Monte Carlo (VBMC) framework, that is variational inference, Gaussian processes and adaptive Bayesian quadrature. The material presented here is largely based and expands on the "theoretical background" section of [1].

## A.1   Variational inference

Let $\boldsymbol{\theta} \in \mathcal{X} \subseteq \mathbb{R}^D$ be a parameter vector of a model of interest, and $\mathcal{D}$ a dataset. Variational inference is an approximate inference framework in which an intractable posterior $p(\boldsymbol{\theta}|\mathcal{D})$ is approximated by a simpler distribution $q(\boldsymbol{\theta}) \equiv q_{\boldsymbol{\phi}}(\boldsymbol{\theta})$ that belongs to a parametric family indexed by parameter vector $\boldsymbol{\phi}$, such as a multivariate normal or a mixture of Gaussians [2, 3]. Thus, the goal of variational inference is to find $\boldsymbol{\phi}$ for which the variational posterior $q_{\boldsymbol{\phi}}$ is "closest" in approximation to the true posterior, according to some measure of discrepancy.

In variational Bayes, the discrepancy between approximate and true posterior is quantified by the Kullback-Leibler (KL) divergence,

$$D_{\mathrm{KL}}\left[q_{\boldsymbol{\phi}}(\boldsymbol{\theta})||p(\boldsymbol{\theta}|\mathcal{D})\right] = \mathbb{E}_{\boldsymbol{\phi}}\left[\log\frac{q_{\boldsymbol{\phi}}(\boldsymbol{\theta})}{p(\boldsymbol{\theta}|\mathcal{D})}\right], \tag{S1}$$

where we adopted the compact notation $\mathbb{E}_{\boldsymbol{\phi}} \equiv \mathbb{E}_{q_{\boldsymbol{\phi}}}$. Crucially, $D_{\mathrm{KL}}(q||p) \geq 0$ and the equality is achieved if and only if $q \equiv p$. $D_{\mathrm{KL}}$ is not symmetric, and the specific choice of using $D_{\mathrm{KL}}\left[q||p\right]$ (*reverse* $D_{\mathrm{KL}}$) as opposed to $D_{\mathrm{KL}}\left[p||q\right]$ (*forward* $D_{\mathrm{KL}}$) is a key feature of the variational framework.

The variational approach casts Bayesian inference as an optimization problem, which consists of finding the variational parameter vector $\boldsymbol{\phi}$ that minimizes Eq. S1. We can rewrite Eq. S1 as

$$\log p(\mathcal{D}) = D_{\mathrm{KL}}\left[q_{\boldsymbol{\phi}}(\boldsymbol{\theta})||p(\boldsymbol{\theta}|\mathcal{D})\right] + \mathcal{F}[q_{\boldsymbol{\phi}}], \tag{S2}$$

where on the left-hand side we have the *model evidence*, and on the right-hand side the KL divergence plus the *negative free energy*, defined as

$$\mathcal{F}\left[q_{\boldsymbol{\phi}}\right] = \mathbb{E}_{\boldsymbol{\phi}}\left[\log\frac{p(\mathcal{D}|\boldsymbol{\theta})p(\boldsymbol{\theta})}{q_{\boldsymbol{\phi}}(\boldsymbol{\theta})}\right] = \mathbb{E}_{\boldsymbol{\phi}}\left[f(\boldsymbol{\theta})\right] + \mathcal{H}[q_{\boldsymbol{\phi}}(\boldsymbol{\theta})], \tag{S3}$$

with $f(\boldsymbol{\theta}) \equiv \log p(\mathcal{D}|\boldsymbol{\theta})p(\boldsymbol{\theta}) = \log p(\mathcal{D}, \boldsymbol{\theta})$ the log joint probability, and $\mathcal{H}[q]$ the entropy of $q$. Now, since as mentioned above the KL divergence is a non-negative quantity, from Eq. S2 we have $\mathcal{F}[q] \leq \log p(\mathcal{D})$, with equality holding if $q(\boldsymbol{\theta}) \equiv p(\boldsymbol{\theta}|\mathcal{D})$. For this reason, Eq. S3 is known as the *evidence lower bound* (ELBO), so called because it is a lower bound to the log marginal likelihood or model evidence. Importantly, maximization of the variational objective, Eq. S3, is equivalent to minimization of the KL divergence, and produces both an approximation of the posterior $q_{\boldsymbol{\phi}}$ and the ELBO, which can be used as a metric for model selection.

Classically, $q$ is chosen to belong to a family (e.g., a factorized posterior, or mean field) such that both the expected log joint in Eq. S3 and the entropy afford analytical solutions, which are then used to yield closed-form equations for a coordinate ascent algorithm. In the VBMC framework, instead, $f(\boldsymbol{\theta})$ is assumed to be a potentially expensive black-box function, which prevents a direct computation of Eq. S3 analytically or via simple numerical integration.

## A.2   Gaussian processes

Gaussian processes (GPs) are a flexible class of statistical models for specifying prior distributions over unknown functions $f : \mathcal{X} \subseteq \mathbb{R}^D \to \mathbb{R}$ [4]. GPs are defined by a mean function $m : \mathcal{X} \to \mathbb{R}$ and

a positive definite covariance, or kernel function $\kappa : \mathcal{X} \times \mathcal{X} \to \mathbb{R}$. VBMC uses the common squared exponential (rescaled Gaussian) kernel,

$$\kappa(\boldsymbol{\theta}, \boldsymbol{\theta}') = \sigma_f^2 \Lambda \mathcal{N}\left(\boldsymbol{\theta}; \boldsymbol{\theta}', \boldsymbol{\Sigma}_\ell\right) \qquad \text{with } \boldsymbol{\Sigma}_\ell = \text{diag}\left[\ell^{(1)^2}, \dots, \ell^{(D)^2}\right], \tag{S4}$$

where $\sigma_f$ is the output length scale, $\boldsymbol{\ell}$ is the vector of input length scales, and $\Lambda \equiv (2\pi)^{\frac{D}{2}} \prod_{i=1}^{D} \ell^{(i)}$ is equal to the normalization factor of the Gaussian (this notation makes it easy to apply Gaussian identities used in Bayesian quadrature). As a mean function, VBMC uses a *negative quadratic* function to ensure well-posedness of the variational formulation, and defined as [1, 5]

$$m(\boldsymbol{\theta}) \equiv m_0 - \frac{1}{2} \sum_{i=1}^{D} \frac{\left(\theta^{(i)} - \theta_{\text{m}}^{(i)}\right)^2}{\omega^{(i)^2}}, \tag{S5}$$

where $m_0$ denotes the maximum, $\boldsymbol{\theta}_{\text{m}}$ is the location, and $\boldsymbol{\omega}$ is a vector of length scales. Finally, GPs are also characterized by a likelihood or observation noise model, which is assumed here to be Gaussian with known variance $\sigma_{\text{obs}}^2(\boldsymbol{\theta})$ for each point in the training set (in the original formulation of VBMC, observation noise is assumed to be a small positive constant).

Conditioned on training inputs $\boldsymbol{\Theta} = \{\boldsymbol{\theta}_1, \dots, \boldsymbol{\theta}_N\}$, observed function values $\boldsymbol{y} = f(\boldsymbol{\Theta})$ and observation noise $\sigma_{\text{obs}}^2(\boldsymbol{\Theta})$, the posterior GP mean and covariance are available in closed form [4],

$$\overline{f}_{\boldsymbol{\Xi}}(\boldsymbol{\theta}) \equiv \mathbb{E}\left[f(\boldsymbol{\theta}) | \boldsymbol{\Xi}, \boldsymbol{\psi}\right] = \kappa(\boldsymbol{\theta}, \boldsymbol{\Theta})\left[\kappa(\boldsymbol{\Theta}, \boldsymbol{\Theta}) + \boldsymbol{\Sigma}_{\text{obs}}(\boldsymbol{\Theta})\right]^{-1}\left(\boldsymbol{y} - m(\boldsymbol{\Theta})\right) + m(\boldsymbol{\theta})$$

$$C_{\boldsymbol{\Xi}}(\boldsymbol{\theta}, \boldsymbol{\theta}') \equiv \text{Cov}\left[f(\boldsymbol{\theta}), f(\boldsymbol{\theta}') | \boldsymbol{\Xi}, \boldsymbol{\psi}\right] = \kappa(\boldsymbol{\theta}, \boldsymbol{\theta}') - \kappa(\boldsymbol{\theta}, \boldsymbol{\Theta})\left[\kappa(\boldsymbol{\Theta}, \boldsymbol{\Theta}) + \boldsymbol{\Sigma}_{\text{obs}}(\boldsymbol{\Theta})\right]^{-1} \kappa(\boldsymbol{\Theta}, \boldsymbol{\theta}'), \tag{S6}$$

where $\boldsymbol{\Xi} = \{\boldsymbol{\Theta}, \boldsymbol{y}, \boldsymbol{\sigma}_{\text{obs}}\}$ is the set of training function data for the GP; $\boldsymbol{\psi}$ is a hyperparameter vector for the GP mean, covariance, and likelihood; and $\boldsymbol{\Sigma}_{\text{obs}}(\boldsymbol{\Theta}) \equiv \text{diag}\left[\sigma_{\text{obs}}^2(\boldsymbol{\theta}_1), \dots, \sigma_{\text{obs}}^2(\boldsymbol{\theta}_N)\right]$ is the observation noise (diagonal) matrix.

### A.3  Adaptive Bayesian quadrature

Bayesian quadrature, also known as cubature when dealing with multi-dimensional integrals, is a technique to obtain Bayesian estimates of intractable integrals of the form [6, 7]

$$Z = \int_{\mathcal{X}} f(\boldsymbol{\theta})\pi(\boldsymbol{\theta})d\boldsymbol{\theta}, \tag{S7}$$

where $f$ is a function of interest and $\pi$ a known probability distribution. For the purpose of VBMC, we consider the domain of integration $\mathcal{X} = \mathbb{R}^D$. When a GP prior is specified for $f$, since integration is a linear operator, the integral $Z$ is also a Gaussian random variable whose posterior mean and variance are [7]

$$\mathbb{E}_{f|\boldsymbol{\Xi}}[Z] = \int \overline{f}_{\boldsymbol{\Xi}}(\boldsymbol{\theta})\pi(\boldsymbol{\theta})d\boldsymbol{\theta}, \qquad \mathbb{V}_{f|\boldsymbol{\Xi}}[Z] = \int \int C_{\boldsymbol{\Xi}}(\boldsymbol{\theta}, \boldsymbol{\theta}')\pi(\boldsymbol{\theta})\pi(\boldsymbol{\theta}')d\boldsymbol{\theta}d\boldsymbol{\theta}'. \tag{S8}$$

Importantly, if $f$ has a Gaussian kernel and $\pi$ is a Gaussian or mixture of Gaussians (among other functional forms), the integrals in Eq. S8 have closed-form solutions.

**Active sampling**  The point $\boldsymbol{\theta}_\star \in \mathcal{X}$ to evaluate next to improve our estimate of the integral (Eq. S7) is chosen via a proxy optimization of a given *acquisition function* $a : \mathcal{X} \to \mathbb{R}$, that is $\boldsymbol{\theta}_\star = \text{argmax}_{\boldsymbol{\theta}} a(\boldsymbol{\theta})$. Previously introduced acquisition functions for Bayesian quadrature include the *expected entropy*, which minimizes the expected entropy of the integral after adding $\boldsymbol{\theta}_\star$ to the training set [8], and a family of strategies under the name of *uncertainty sampling*, whose goal is generally to find the point with maximal (pointwise) variance of the *integrand* at $\boldsymbol{\theta}_\star$ [9]. The standard acquisition function for VBMC is *prospective uncertainty sampling* (see main text and [1, 5]). Recent work proved convergence guarantees for active-sampling Bayesian quadrature under a broad class of acquisition functions which includes various forms of uncertainty sampling [10].

## B  Algorithmic details

We report here implementation details of new or improved features of the VBMC algorithm omitted from the main text.

## B.1 Modified VBMC features

In this section, we describe minor changes to the basic VBMC framework. For implementation details of the algorithm which have remained unchanged, we refer the reader to the main text and Supplement of the original VBMC paper [1].

**Reliability index**   In VBMC, the *reliability index* $r(t)$ is a metric computed at the end of each iteration $t$ and determines, among other things, the termination condition [1]. We recall that $r(t)$ is computed as the arithmetic mean of three reliability features:

1. The absolute change in mean ELBO from the previous iteration: $r_1(t) = |\mathbb{E}[\text{ELBO}(t)] - \mathbb{E}[\text{ELBO}(t-1)]| / \Delta_{\text{SD}}$.
2. The uncertainty of the current ELBO: $r_2(t) = \sqrt{\mathbb{V}[\text{ELBO}(t)]} / \Delta_{\text{SD}}$.
3. The change in 'Gaussianized' symmetrized KL divergence (see Eq. S21) between the current and previous-iteration variational posterior $q_t \equiv q_{\phi_t}(\boldsymbol{\theta})$: $r(t) = \text{gsKL}(q_t || q_{t-1}) / \Delta_{\text{KL}}$.

The parameters $\Delta_{\text{SD}}$ and $\Delta_{\text{KL}}$ are tolerance hyperparameters, chosen such that $r_j \lesssim 1$, with $j = 1, 2, 3$, for features that are deemed indicative of a good solution. We set $\Delta_{\text{KL}} = 0.01 \cdot \sqrt{D}$ as in the original VBMC paper. To account for noisy observations, we set $\Delta_{\text{SD}}$ in the current iteration equal to the geometric mean between the baseline $\Delta_{\text{SD}}^{\text{base}} = 0.1$ (from the original VBMC paper) and the GP noise in the high-posterior density region, $\sigma_{\text{obs}}^{\text{hpd}}$, and constrain it to be in the $[0.1, 1]$ range. That is,

$$\Delta_{\text{SD}} = \min\left[1, \max\left[0.1, \sqrt{\Delta_{\text{SD}}^{\text{base}} \cdot \sigma_{\text{obs}}^{\text{hpd}}}\right]\right], \tag{S9}$$

where $\sigma_{\text{obs}}^{\text{hpd}}$ is computed as the median observation noise at the top 20% points in terms of log-posterior value in the GP training set.

**Regularization of acquisition functions**   In VBMC, active sampling is performed by maximizing a chosen acquisition function $a : \mathcal{X} \subseteq \mathbb{R}^D \to [0, \infty)$, where $\mathcal{X}$ is the support of the target density (see Section C). In practice, in VBMC we maximize a *regularized* acquisition function

$$a^{\text{reg}}(\boldsymbol{\theta}; a) \equiv a(\boldsymbol{\theta}) b_{\text{var}}(\boldsymbol{\theta}) b_{\text{bnd}}(x) \tag{S10}$$

where $b_{\text{var}}(\boldsymbol{\theta})$ is a GP variance regularization term introduced in [1],

$$b_{\text{var}}(\boldsymbol{\theta}) = \exp\left\{-\left(\frac{V^{\text{reg}}}{V_{\Xi}(\boldsymbol{\theta})} - 1\right) |[V_{\Xi}(\boldsymbol{\theta}) < V^{\text{reg}}]|\right\} \tag{S11}$$

where $V_{\Xi}(\boldsymbol{\theta})$ is the posterior latent variance of the GP, $V^{\text{reg}}$ a regularization parameter (we use $V^{\text{reg}} = 10^{-4}$), and we denote with $|[\cdot]|$ *Iverson's bracket* [11], which takes value 1 if the expression inside the bracket is true, 0 otherwise. Eq. S11 penalizes the selection of points too close to an existing input, which might produce numerical issues.

The $b_{\text{bnd}}$ term is a new term that we added in this work to discard points too close to the parameter bounds, which would map to very large positive or negative values in the unbounded inference space,

$$b_{\text{bnd}}(\boldsymbol{\theta}) = \begin{cases} 1 & \text{if } \tilde{\theta}^{(i)} \geq \text{LB}_{\varepsilon}^{(i)} \wedge \tilde{\theta}^{(i)} \leq \text{UB}_{\varepsilon}^{(i)}, \text{ for all } 1 \leq i \leq D \\ 0 & \text{otherwise} \end{cases} \tag{S12}$$

where $\tilde{\boldsymbol{\theta}}(\boldsymbol{\theta})$ is the parameter vector remapped to the original space, and $\text{LB}_{\varepsilon}^{(i)} \equiv \text{LB}^{(i)} + \varepsilon(\text{UB}^{(i)} - \text{LB}^{(i)})$, $\text{UB}_{\varepsilon}^{(i)} \equiv \text{UB}^{(i)} - \varepsilon(\text{UB}^{(i)} - \text{LB}^{(i)})$, with $\varepsilon = 10^{-5}$.

**GP hyperparameters and priors**   The GP model in VBMC has $3D + 3$ hyperparameters, $\boldsymbol{\psi} = (\boldsymbol{\ell}, \sigma_f, \overline{\sigma}_{\text{obs}}, m_0, \boldsymbol{\theta}_{\text{m}}, \boldsymbol{\omega})$. All scale hyperparameters, that is $\{\boldsymbol{\ell}, \sigma_f, \overline{\sigma}_{\text{obs}}, \boldsymbol{\omega}\}$, are defined in log space. Each hyperparameter has an independent prior, either bounded uniform or a truncated Student's $t$ distribution with mean $\mu$, scale $\sigma$, and $\nu = 3$ degrees of freedom. GP hyperparameters and their priors are reported in Table S1.

In Table S1, $\boldsymbol{L}$ denotes the vector of plausible ranges along each coordinate dimension, with $L^{(i)} \equiv \text{PUB}^{(i)} - \text{PLB}^{(i)}$. The base observation noise $\overline{\sigma}_{\text{obs}}^2$ is a constant added to the input-dependent observation

| Hyperparameter | Description | Prior mean $\mu$ | Prior scale $\sigma$ |
|---|---|---|---|
| $\log \ell^{(i)}$ | Input length scale | $\log \left[ \sqrt{\frac{D}{6}} L^{(i)} \right]$ | $\log \sqrt{10^3}$ |
| $\log \sigma_f$ | Output scale | Uniform | — |
| $\log \overline{\sigma}_{\mathrm{obs}}$ | Base observation noise | $\log \sqrt{10^{-5}}$ | 0.5 |
| $m_0$ | Mean function maximum | Uniform | — |
| $x_{\mathrm{m}}^{(i)}$ | Mean function location | Uniform | — |
| $\log \omega^{(i)}$ | Mean function scale | Uniform | — |

Table S1: GP hyperparameters and their priors. See text for more information.

noise $\sigma_{\mathrm{obs}}^2(\boldsymbol{\theta})$. Note that we have modified the GP hyperparameter priors with respect to the original VBMC paper, and these are now the default settings for both noisy and noiseless inference. In particular, we removed dependence of the priors from the GP training set (the 'empirical Bayes' approach previously used), as it was found to occasionally generate unstable behavior.

**Frequent retrain** In the original VBMC algorithm, the GP model and variational posterior are re-trained only at the end of each iteration, corresponding to $n_{\mathrm{active}} = 5$ likelihood evaluations. However, in the presence of observation noise, approximation of both the GP and the variational posterior may benefit from a more frequent update. Thus, for noisy likelihoods we introduced a *frequent retrain*, that is fast re-training of both the GP and of the variational posterior within the active sampling loop, after each new function evaluation. This frequent update sets VBMC on par with other algorithms, such as GP-IMIQR and WSABI, which similarly retrain the GP representation after each likelihood evaluation. In VBMC, frequent retrain is active throughout the warm-up stage. After warm-up, we activate frequent retrain only when the previous iteration's reliability index $r(t-1) > 3$, indicating that the solution has not stabilized yet.

## B.2   Variational whitening

We start performing *variational whitening* $\tau_{\mathrm{vw}}$ iterations after the end of warm-up, and then subsequently at increasing intervals of $k\tau_{\mathrm{vw}}$ iterations, where $k$ is the count of previously performed whitenings ($\tau_{\mathrm{vw}} = 5$ in this work). Moreover, variational whitening is postponed until the reliability index $r(t)$ of the current iteration is below 3, indicating a degree of stability of the current variational posterior (see Section B.1). Variational whitening consists of a linear transformation $\mathbf{W}$ of the inference space (a rotation and rescaling) such that the variational posterior $q_{\boldsymbol{\phi}}$ obtains unit diagonal covariance matrix. We compute the covariance matrix $\mathbf{C}_{\boldsymbol{\phi}}$ of $q_{\boldsymbol{\phi}}$ analytically, and we set the entries whose correlation is less than 0.05 in absolute value to zero, yielding a corrected covariance matrix $\widetilde{\mathbf{C}}_{\boldsymbol{\phi}}$. We then calculate the whitening transform $\mathbf{W}$ by performing a singular value decomposition (SVD) of $\widetilde{\mathbf{C}}_{\boldsymbol{\phi}}$.

## C   Acquisition functions

In this Section, we report derivations and additional implementation details for the acquisition functions introduced in the main text.

### C.1   Observation noise

All acquisition functions in the main text require knowledge of the log-likelihood observation noise $\sigma_{\mathrm{obs}}(\boldsymbol{\theta})$ at an arbitrary point $\boldsymbol{\theta} \in \mathcal{X}$. However, we only assumed availability of an estimate $(\widehat{\sigma}_{\mathrm{obs}})_n$ of $\sigma_{\mathrm{obs}}(\boldsymbol{\theta}_n)$ for all parameter values evaluated so far, $1 \le n \le N$. We estimate values of $\sigma_{\mathrm{obs}}(\boldsymbol{\theta})$ outside the training set via a simple nearest-neighbor approximation, that is

$$\sigma_{\mathrm{obs}}(\boldsymbol{\theta}_\star) = \sigma_{\mathrm{obs}}(\boldsymbol{\theta}_n) \quad \text{for } n = \arg \min_{1 \le n \le N} d_{\boldsymbol{\ell}}(\boldsymbol{\theta}_\star, \boldsymbol{\theta}_n), \tag{S13}$$

where $d_{\boldsymbol{\ell}}$ is the rescaled Euclidean distance between two points in inference space, where each coordinate dimension $i$ has been rescaled by the GP input length $\ell_i$, with $1 \le i \le D$. When multiple GP hyperparameter samples are available, we use the geometric mean of each input length across samples. Eq. S13 may seem like a coarse approximation, but we found it effective in practice.

## C.2 Expected information gain (EIG)

The *expected information gain* (EIG) acquisition function $a_{\text{EIG}}$ is based on a mutual information maximizing acquisition function for Bayesian quadrature introduced in [12].

First, note that the *information gain* is defined as the KL-divergence between posterior and prior; in our case, between the posterior of the log joint $\mathcal{G}$ after observing value $y_\star$ at $\boldsymbol{\theta}_\star$, and the current posterior over $\mathcal{G}$ given the observed points in the training set, $\boldsymbol{\Xi} = \{\boldsymbol{\Theta}, \boldsymbol{y}, \boldsymbol{\sigma}_{\text{obs}}\}$. Since $y_\star$ is yet to be observed, we consider then the *expected* information gain of performing a measurement at $\boldsymbol{\theta}_\star$, that is

$$\text{EIG}(\boldsymbol{\theta}_\star; \boldsymbol{\Xi}_t) = \mathbb{E}_{y_\star|\boldsymbol{\theta}_\star}\left[D_{\text{KL}}\left(p(\mathcal{G}|\boldsymbol{\Xi} \cup \{(\boldsymbol{\theta}_\star, y_\star, \sigma_{\text{obs}\star})\}) \,\|\, p(\mathcal{G}|\boldsymbol{\Xi})\right)\right]. \tag{S14}$$

It can be shown that Eq. S14 is identical to the *mutual information* between $\mathcal{G}$ and $\boldsymbol{y}_\star$ [13]

$$I\left[\mathcal{G}; y_\star\right] = H[\mathcal{G}] + H[y_\star] - H[\mathcal{G}, y_\star] \tag{S15}$$

where $H(\cdot)$ denotes the (marginal) differential entropy and $H(\cdot, \cdot)$ the joint entropy. By the definition of GP, $y_\star$ is normally distributed, and so is each component $\mathcal{G}_k$ of the log-joint, due to Bayesian quadrature (see Section A). As a weighted sum of normally distributed random variables, $\mathcal{G}$ is also normally distributed, and so is the joint distribution of $y_\star$ and $\mathcal{G}$. We recall that the differential entropy of a bivariate normal distribution with covariance matrix $\mathbf{A} \in \mathbb{R}^{2\times2}$ is $H = \log(2\pi e) + \frac{1}{2}\log|\mathbf{A}|$. Thus we have (see Eq. 7 in the main text)

$$a_{\text{EIG}}(\boldsymbol{\theta}_\star) \equiv I\left[\mathcal{G}; y_\star\right] = -\frac{1}{2}\log\left(1 - \rho^2(\boldsymbol{\theta}_\star)\right), \qquad \text{with } \rho(\boldsymbol{\theta}_\star) \equiv \frac{\mathbb{E}_\phi\left[C_{\boldsymbol{\Xi}}(f(\cdot), f(\boldsymbol{\theta}_\star))\right]}{\sqrt{v_{\boldsymbol{\Xi}}(\boldsymbol{\theta}_\star)\mathbb{V}_{f|\boldsymbol{\Xi}}[\mathcal{G}]}}, \tag{S16}$$

where we used the *scalar correlation* $\rho(\cdot)$ [12]; and $C_{\boldsymbol{\Xi}}(\cdot, \cdot)$ is the GP posterior covariance, $v_{\boldsymbol{\Xi}}(\cdot)$ the GP posterior predictive variance (including observation noise), and $\mathbb{V}_{f|\boldsymbol{\Xi}}[\mathcal{G}]$ the posterior variance of the expected log joint – all given the current training set $\boldsymbol{\Xi}$.

The expected value at the numerator of $\rho(\boldsymbol{\theta}_\star)$ is

$$\begin{aligned}
\mathbb{E}_\phi\left[C_{\boldsymbol{\Xi}}(f(\cdot), f(\boldsymbol{\theta}_\star))\right] &= \int q(\boldsymbol{\theta})C_{\boldsymbol{\Xi}}\left(f(\boldsymbol{\theta}), f(\boldsymbol{\theta}_\star)\right) d\boldsymbol{\theta} \\
&= \sum_{k=1}^K w_k \int \mathcal{N}\left(\boldsymbol{\theta}; \boldsymbol{\mu}_k, \sigma_k^2\boldsymbol{\Sigma}\right) C_{\boldsymbol{\Xi}}\left(f(\boldsymbol{\theta}), f(\boldsymbol{\theta}_\star)\right) d\boldsymbol{\theta} \\
&= \sum_{k=1}^K w_k \mathcal{K}_k(\boldsymbol{\theta}_\star),
\end{aligned} \tag{S17}$$

where we recall that $w_k$, $\boldsymbol{\mu}_k$, and $\sigma_k$ are, respectively, the mixture weight, mean, and scale of the $k$-th component of the variational posterior $q$, for $1 \leq k \leq K$; $\boldsymbol{\Sigma}$ is a common diagonal covariance matrix $\boldsymbol{\Sigma} \equiv \text{diag}[\lambda^{(1)^2}, \ldots, \lambda^{(D)^2}]$; and $C_{\boldsymbol{\Xi}}$ is the GP posterior covariance as per Eq. S6. Finally, each term in Eq. S17 can be written as

$$\begin{aligned}
\mathcal{K}_k(\boldsymbol{\theta}_\star) &= \int \mathcal{N}\left(\boldsymbol{\theta}; \boldsymbol{\mu}_k, \sigma_k^2\boldsymbol{\Sigma}\right)\left[\sigma_f^2\Lambda\mathcal{N}\left(\boldsymbol{\theta}; \boldsymbol{\theta}_\star, \boldsymbol{\Sigma}_\ell\right) \ldots \right. \\
&\qquad \left. \ldots - \sigma_f^2\Lambda\mathcal{N}\left(\boldsymbol{\theta}; \boldsymbol{\Theta}, \boldsymbol{\Sigma}_\ell\right)\left[\kappa(\boldsymbol{\Theta}, \boldsymbol{\Theta}) + \boldsymbol{\Sigma}_{\text{obs}}(\boldsymbol{\Theta})\right]^{-1}\sigma_f^2\Lambda\mathcal{N}\left(\boldsymbol{\Theta}; \boldsymbol{\theta}_\star, \boldsymbol{\Sigma}_\ell\right)\right]d\boldsymbol{\theta} \\
&= \sigma_f^2\Lambda\mathcal{N}\left(\boldsymbol{\theta}_\star; \boldsymbol{\mu}_k, \boldsymbol{\Sigma}_\ell + \sigma_k^2\boldsymbol{\Sigma}\right) - \sigma_f^2\Lambda\boldsymbol{z}_k^\top\left[\kappa(\boldsymbol{\Theta}, \boldsymbol{\Theta}) + \boldsymbol{\Sigma}_{\text{obs}}(\boldsymbol{\Theta})\right]^{-1}\mathcal{N}\left(\boldsymbol{\Theta}; \boldsymbol{\theta}_\star, \boldsymbol{\Sigma}_\ell\right),
\end{aligned} \tag{S18}$$

where $\boldsymbol{z}_k$ is a $N$-dimensional vector with entries $z_k^{(n)} = \sigma_f^2\Lambda\mathcal{N}\left(\boldsymbol{\mu}_k; \boldsymbol{\theta}_n, \sigma_k^2\boldsymbol{\Sigma} + \boldsymbol{\Sigma}_\ell\right)$ for $1 \leq n \leq N$.

## C.3 Integrated median / variational interquantile range (IMIQR/ VIQR)

The *integrated median interquantile range* (IMIQR) acquisition function has been recently proposed in [14] as a robust, principled metric for posterior estimation with noisy evaluations (see also Eq. 8 in the main text),

$$a_{\text{IMIQR}}(\boldsymbol{\theta}_\star) = -2\int_{\mathcal{X}} \exp\left(\overline{f}_{\boldsymbol{\Xi}}(\boldsymbol{\theta})\right)\sinh\left(us_{\boldsymbol{\Xi}\cup\boldsymbol{\theta}_\star}(\boldsymbol{\theta})\right)d\boldsymbol{\theta}. \tag{S19}$$

It combines two ideas: (a) using the interquantile range (IQR) as a robust measure of uncertainty, as opposed to the variance; and (b) approximating the median integrated IQR loss, which follows

from decision-theoretic principles but is intractable, with the integrated median IQR, which can be computed somewhat more easily [14]. Note that Eq. S19 differs slightly from Eq. 30 in [14] in that in our definition the prior term is subsumed into the joint distribution, with no loss of generality.

A major issue with Eq. S19 is that the integral is still intractable. By noting that $\exp\left(\overline{f}_{\boldsymbol{\Xi}}(\boldsymbol{\theta})\right)$ is the joint distribution, in VBMC we can replace it with the variational posterior, obtaining thus the *variational* (integrated median) *interquantile range* acquisition function $a_{\text{VIQR}}$ (see main text).

## D  Benchmark details

We report here details about the benchmark setup, in particular parameter bounds and dataset information for all problems in the benchmark (Section D.1); how we adapted the WSABI and GP-IMIQR algorithms for the purpose of our noisy benchmark (Section D.2); and the computing infrastructure (Section D.3).

### D.1  Problem specification

**Parameter bounds**   We report in Table S2 the parameter bounds used in the problems of the noisy-inference benchmark. We denote with LB and UB the hard lower and upper bounds, respectively; whereas with PLB and PUB we denote the 'plausible' lower and upper bounds, respectively [1, 15]. Plausible ranges should identify a region of high posterior probability mass in parameter space given our knowledge of the model and of the data; lacking other information, these are recommended to be set to e.g. the $\sim 68\%$ high-density interval according to the marginal prior probability in each dimension [1]. Plausible values are used to initialize and set hyperparameters of some of the algorithms. For example, the initial design for VBMC and GP-IMIQR is drawn from a uniform distribution over the plausible box in inference space.

**Dataset information**

- **Ricker:** We generated a synthetic dataset of $T = 50$ observations using the "true" parameter vector $\boldsymbol{\theta}_{\text{true}} = (3.8, 10, 0.3)$ with $T = 50$, as in [14].

- **Attentional drift-diffusion model (aDDM):** We used fixation and choice data from two participants (subject #13 and subject #16 from [16]) who completed all $N = 100$ trials in the experiment without technical issues (reported as 'missing trials' in the data).

- **Bayesian timing:** We analyzed reproduced time intervals of one representative subject from Experiment 3 (uniform distribution; subject #2) in [17], with $N = 1512$ trials.

- **Multisensory causal inference:** We examined datasets of subject #1 and #2 from the explicit causal inference task ('unity judgment') in [18]; with respectively $N = 1069$ and $N = 857$ trials, across three different visual coherence conditions.

- **Neuronal selectivity:** We analyzed two neurons (one from area V1, one from area V2 of primate visual cortex) from [19], both with $N = 1760$ trials. The same datasets have been used in previous optimization and inference benchmarks [1, 5, 15].

- **Rodent 2AFC:** We took a representative rat subject from [20], already used for demonstration purposes by [21], limiting our analysis of choice behavior to the last $N = 10^4$ trials in the data set.

### D.2  Algorithm specification

**WSABI**   Warped sequential active Bayesian integration (WSABI) is a technique to compute the log marginal likelihood via GP surrogate models and Bayesian quadrature [9]. In this work, we use WSABI as an example of a surrogate-based method for model evidence approximation different from VBMC. The key idea of WSABI is to model directly the *square root* of the likelihood function $\mathcal{L}$ (as opposed to the log-likelihood) via a GP,

$$\tilde{\mathcal{L}}(\boldsymbol{\theta}) \equiv \sqrt{2\left(\mathcal{L}(\boldsymbol{\theta}) - \alpha\right)} \quad \implies \quad \mathcal{L}(\boldsymbol{\theta}) = \alpha + \frac{1}{2}\tilde{\mathcal{L}}(\boldsymbol{\theta})^2, \tag{S20}$$

Table S2: Parameters and bounds of all models (before remapping to inference space).

| Model | Parameter | Description | LB | UB | PLB | PUB |
|---|---|---|---|---|---|---|
| Ricker | $\log(r)$ | Growth factor (log) | 3 | 5 | 3.2 | 4.8 |
| | $\phi$ | Observed fraction | 4 | 20 | 5.6 | 18.4 |
| | $\sigma_\varepsilon$ | Growth noise | 0 | 0.8 | 0.08 | 0.72 |
| aDDM | $d$ | Drift rate | 0 | 5 | 0.1 | 2 |
| | $\beta$ | Attentional bias factor | 0 | 1 | 0.1 | 0.9 |
| | $\sigma_\varepsilon$ | Diffusion noise | 0.1 | 2 | 0.2 | 1 |
| | $\lambda$ | Lapse rate | 0.01 | 0.2 | 0.03 | 0.1 |
| Bayesian timing | $w_s$ | Sensory noise (Weber's fraction) | 0.01 | 0.5 | 0.05 | 0.25 |
| | $w_m$ | Motor noise (Weber's fraction) | 0.01 | 0.5 | 0.05 | 0.25 |
| | $\mu_p$ | Prior mean (seconds) | 0.3 | 1.95 | 0.6 | 0.975 |
| | $\sigma_p$ | Prior standard deviation (seconds) | 0.0375 | 0.75 | 0.075 | 0.375 |
| | $\lambda$ | Lapse rate | 0.01 | 0.2 | 0.02 | 0.05 |
| Multisensory causal inference (CI) | $\sigma_{\text{vest}}$ | Vestibular noise (deg) | 0.5 | 80 | 1 | 40 |
| | $\sigma_{\text{vis}}(c_{\text{low}})$ | Visual noise, low coherence (deg) | 0.5 | 80 | 1 | 40 |
| | $\sigma_{\text{vis}}(c_{\text{med}})$ | Visual noise, medium coherence (deg) | 0.5 | 80 | 1 | 40 |
| | $\sigma_{\text{vis}}(c_{\text{high}})$ | Visual noise, high coherence (deg) | 0.5 | 80 | 1 | 40 |
| | $\kappa$ | 'Sameness' threshold (deg) | 0.25 | 180 | 1 | 45 |
| | $\lambda$ | Lapse rate | 0.005 | 0.5 | 0.01 | 0.2 |
| Neuronal selectivity | $\theta_1$ | Preferred direction of motion (deg) | 0 | 360 | 90 | 270 |
| | $\theta_2$ | Preferred spatial freq. (cycles/deg) | 0.05 | 15 | 0.5 | 10 |
| | $\theta_3$ | Aspect ratio of 2-D Gaussian | 0.1 | 3.5 | 0.3 | 3.2 |
| | $\theta_4$ | Derivative order in space | 0.1 | 3.5 | 0.3 | 3.2 |
| | $\theta_5$ | Gain inhibitory channel | -1 | 1 | -0.3 | 0.3 |
| | $\theta_6$ | Response exponent | 1 | 6.5 | 1.01 | 5 |
| | $\theta_7$ | Variance of response gain | 0.001 | 10 | 0.015 | 1 |
| Rodent 2AFC | $w_0$ | Bias | -3 | 3 | -1 | 1 |
| | $w_c$ | Weight on 'previous correct side' | -3 | 3 | -1 | 1 |
| | $w_{\overline{s}}$ | Weight on long-term history | -3 | 3 | -1 | 1 |
| | $w_L^{(0)}$ | Weight on left stimulus ($t=0$) | -3 | 3 | -1 | 1 |
| | $w_L^{(-1)}$ | Weight on left stimulus ($t=-1$) | -3 | 3 | -1 | 1 |
| | $w_L^{(-2)}$ | Weight on left stimulus ($t=-2$) | -3 | 3 | -1 | 1 |
| | $w_R^{(0)}$ | Weight on right stimulus ($t=0$) | -3 | 3 | -1 | 1 |
| | $w_R^{(-1)}$ | Weight on right stimulus ($t=-1$) | -3 | 3 | -1 | 1 |
| | $w_R^{(-2)}$ | Weight on right stimulus ($t=-2$) | -3 | 3 | -1 | 1 |

where $\alpha$ is a small positive scalar. If $\tilde{\mathcal{L}}$ is modeled as a GP, $\mathcal{L}$ is not itself a GP (right-hand side of Eq. S20), but it can be approximated as a GP via a linearization procedure (WSABI-L in [9]), which is the approach we follow throughout our work.

The WSABI algorithm requires an unlimited inference space $\mathcal{X} \equiv \mathbb{R}^D$ and a multivariate normal prior [9]. In our benchmark, all parameters have bound constraints, so we first map the original space to an unbounded inference space via a rescaled logit transform, with an appropriate log-Jacobian correction to the log posterior (see e.g., [1, 22]). Also, in our benchmark all priors are assumed to be uniform. Thus, we pass to WSABI a 'pseudo-prior' consisting of a multivariate normal centered on the middle of the *plausible box*, and with standard deviations equal to half the *plausible range* in each coordinate direction in inference space (see Section D.1). We then correct for this added pseudo-prior by subtracting the log-pseudo-prior value from each log-joint evaluation.

**WSABI with noisy likelihoods**    The original WSABI algorithm does not explicitly support observation noise in the (log-)likelihood. Thus, we modified WSABI to include noisy likelihood evaluations, by mapping noise on the log-likelihood to noise in the square-root likelihood via an unscented trans-

form, and by modifying WSABI's uncertainty-sampling acquisition function to account for observation noise (similarly to what we did for $a_{\mathrm{npro}}$, see main text). However, we found the noise-adjusted WSABI to perform abysmally, even worse than the original WSABI on our noisy benchmark. This failure is likely due to the particular representation used by WSABI (Eq. S20). Crucially, even moderate noise on the log-likelihood translates to extremely large noise on the (square-root) likelihood. Due to this large observation noise, the latent GP will revert to the GP mean function, which corresponds to the constant $\alpha$ (Eq. S20). In the presence of modeled log-likelihood noise, thus, the GP representation of WSABI becomes near-constant and practically useless. For this reason, here and in the main text we report the results of WSABI *without* explicitly added support for observation noise. More work is needed to find an alternative representation of WSABI which would not suffer from observation noise, but it is beyond the scope of our paper.

**GP-IMIQR**  For the GP-IMIQR algorithm described in [14], we used the latest implementation (V3) publicly available at: `https://github.com/mjarvenpaa/parallel-GP-SL`. We considered the IMIQR acquisition function with sequential sampling strategy; the best-performing acquisition function in the empirical analyses in [14]. We used the code essentially 'as is', with minimal changes to interface the algorithm to our noisy benchmark. We ran the algorithm with the recommended default hyperparameters. Given the particularly poor performance of GP-IMIQR on some problems (e.g., Timing, Neuronal), which we potentially attributed to convergence failures of the default MCMC sampling algorithm (DRAM; [23]), we also reran the method with an alternative and robust sampling method (parallel slice sampling; [18, 24]). However, performance of GP-IMIQR with slice sampling was virtually identical, and similarly poor, to its performance with DRAM (data not shown). We note that the same grave issues with the Neuronal model emerged even when we forced initialization of the algorithm in close vicinity of the mode of the posterior (data not shown). We attribute the inability of GP-IMIQR to make significant progress on some problems to excessive exploration, which may lead to GP instabilities; although further investigation is needed to identify the exact causes, beyond the scope of this work.

### D.3  Computing infrastructure

All benchmark runs were performed on MATLAB 2017a (Mathworks, Inc.) using a High Performance Computing cluster whose details can be found at the following link: `https://wikis.nyu.edu/display/NYUHPC/Clusters+-+Prince`. Since different runs may have been assigned to compute nodes with vastly different loads or hardware, we regularly assessed execution speed by performing a set of basic speed benchmark operations (`bench` in MATLAB; considering only numerical tasks). Running times were then converted to the estimated running time on a reference machine, a laptop computer with 16.0 GB RAM and Intel(R) Core(TM) i7-6700HQ CPU @ 2.60 GHz, forced to run single-core during the speed test.

## E  Additional results

We include here a series of additional experimental results and plots omitted from the main text for reasons of space. First, we report the results of the posterior inference benchmark with a different metric (Section E.1). Then, we present results of a robustness analysis of solutions across runs (Section E.2) and of an ablation study (Section E.3). In Section E.4, we show a comparison of true and approximate posteriors for all problems in the benchmark. Then, we study sensitivity of VBMC-VIQR to imprecision in the log-likelihood noise estimates (Section E.5). Finally, we report results for an additional synthetic problem, the g-and-k model (Section E.6).

### E.1  Gaussianized symmetrized KL divergence (gsKL) metric

In the main text, we measured the quality of the posterior approximation via the mean marginal total variation distance (MMTV) between true and appoximate posteriors, which quantifies the distance between posterior marginals. Here we consider an alternative loss metric, the "Gaussianized" symmetrized Kullback-Leibler divergence (gsKL), which is sensitive instead to differences in means and covariances [1]. Specifically, the gsKL between two pdfs $p$ and $q$ is defined as

$$\mathrm{gsKL}(p, q) = \frac{1}{2} \left[ D_{\mathrm{KL}} \left( \mathcal{N}[p] || \mathcal{N}[q] \right) + D_{\mathrm{KL}} (\mathcal{N}[q] || \mathcal{N}[p]) \right], \tag{S21}$$

where $\mathcal{N}[p]$ is a multivariate normal distribution with mean equal to the mean of $p$ and covariance matrix equal to the covariance of $p$ (and same for $q$). Eq. S21 can be expressed in closed form in terms of the means and covariance matrices of $p$ and $q$.

Fig. S1 shows the gsKL between approximate posterior and ground truth, for all algorithms and inference problems considered in the main text. For reference, two Gaussians with unit variance and whose means differ by $\sqrt{2}$ (resp., $\frac{1}{2}$) have a gsKL of 1 (resp., $\frac{1}{8}$). For this reason, we consider a desirable target to be (much) less than 1. Results are qualitatively similar to what we observed for the MMTV metric (Fig. 3 in the main text), in that the ranking and convergence properties of different methods is the same for MMTV and gsKL. In particular, previous state-of-the art method GP-IMIQR fails to converge in several challenging problems (Timing, Neuronal and Rodent); among the variants of VBMC, VBMC-VIQR and VBMC-IMIQR are the only ones that perform consistently well.

Figure S1: **Posterior estimation loss (gsKL).** Median Gaussianized symmetrized KL divergence (gsKL) between the algorithm's posterior and ground truth, as a function of number of likelihood evaluations. A desirable target (dashed line) is less than 1. Shaded areas are 95% CI of the median across 100 runs.

## E.2 Worse-case analysis (90% quantile)

In the main text and other parts of this Supplement, we showed for each performance metric the *median* performance across multiple runs, to convey the 'average-case' performance of an algorithm; in that we expect performance to be at least as good as the median for about half of the runs. To assess the robustness of an algorithm, we are also interested in a 'worse-case' analysis that looks at higher quantiles of the distribution of performance, which are informative of how bad performance can reasonably get (e.g., we expect only about one run out of ten to be worse than the 90% quantile).

We show in Figure S2 the 90% quantile of the MMTV metric, to be compared with Fig. 3 in the main text (results for other metrics are analogous). These results show that the best-performing algorithms, VBMC-VIQR and VBMC-IMIQR, are also the most robust, as both methods manage to achieve good solutions most of the time (with one method working slightly better than the other on some problems, and vice versa). By contrast, other methods such as GP-IMIQR show more variability, in that on some problems (e.g., aDDM) they may have reasonably good median performance, but much higher error when looking at the 90% quantile.

## E.3 Ablation study

We show here the performance of the VBMC algorithm after removing some of the features considered in the main paper. As a baseline algorithm we take VBMC-VIQR. First, we show VBMC-NOWV, obtained by removing from the baseline the 'variational whitening' feature (see main text and Section

Figure S2: **Worse-case posterior estimation loss (MMTV).** 90% quantile of the mean marginal total variation distance (MMTV) between the algorithm's posterior and ground truth, as a function of number of likelihood evaluations. A desirable target (dashed line) is less than 0.2, corresponding to more than 80% overlap between true and approximate posterior marginals (on average across model parameters). Shaded areas are 95% CI of the 90% quantile across 100 runs.

B.2). Second, we consider a variant of VBMC-VIQR in which we do not sample GP hyperparameters from the hyperparameter posterior, but simply obtain a point estimate through maximum-a-posteriori estimation (VBMC-MAP). Optimizing GP hyperparameters, as opposed to a Bayesian treatment of hyperparameters, is a common choice for many surrogate-based methods (e.g., WSABI, GP-IMIQR, although the latter integrates analytically over the GP mean function), so we investigate whether it is needed for VBMC. Finally, we plot the performance of VBMC in its original implementation (VBMC-OLD), as per the VBMC paper [1]. For reference, we also plot both the VBMC-VIQR and GP-IMIQR algorithms, as per Fig. 3 in the main text.

We show in Fig. S3 the results for the MMTV metric, although results are similar for other inference metrics. We can see that all 'lesioned' versions of VBMC perform generally worse than VBMC-VIQR, to different degree, and more visibly in more difficult inference problems. However, for example, VBMC-MAP still performs substantially better than GP-IMIQR, suggesting that the difference in performance between VBMC and GP-IMIQR is not simply because VBMC marginalizes over GP hyperparameters. It is also evident that the previous version of VBMC (VBMC-OLD) is extremely ineffective in the presence of noisy log-likelihoods.

### E.4  Comparison of true and approximate posteriors

We plot in Fig. S4 a comparison between the 'true' marginal posteriors, obtained for all problems via extensive MCMC sampling, and example approximate posteriors recovered by VBMC-VIQR after $50 \times (D + 2)$ likelihood evaluations, the budget allocated for the benchmark. As already quantified by the MMTV metric, we note that VBMC is generally able to obtain good approximations of the true posterior marginals. The effect of noise becomes more prominent when the posteriors are nearly flat, in which case we see greater variability in the VBMC solutions for some parameters (e.g., in the challenging Rodent problem). Note that this is also a consequence of choosing non-informative uniform priors over bounded parameter ranges in our benchmark, which is not necessarily best practice on real problems; (weakly) informative priors should be preferred in most cases [25].

To illustrate the ability of VBMC-VIQR to recover complex interactions in the posterior distribution (and not only univariate marginals) in the presence of noise, we plot in Fig. S5 the full pairwise posterior for one of the problems in the benchmark (Timing model). We can see that the approximate posterior matches the true posterior quite well, with some underestimation of the distribution tails. Underestimation of posterior variance is a common problem for variational approximations [26]

Figure S3: **Lesion study; posterior estimation loss (MMTV).** Median mean marginal total variation distance (MMTV) between the algorithm's posterior and ground truth, as a function of number of likelihood evaluations. Shaded areas are 95% CI of the median across 100 runs.

Figure S4: **True and approximate marginal posteriors.** Each panel shows the ground-truth marginal posterior distribution (red line) for each parameter of problems in the noisy benchmark (rows). For each problem, black lines are marginal distributions of five randomly-chosen approximate posteriors returned by VBMC-VIQR.

and magnified here by the presence of noisy log-likelihood evaluations, and it represents a potential direction of improvement for future work.

Figure S5: **True and approximate posterior of Timing model. A.** Triangle plot of the 'true' posterior (obtained via MCMC) for the Timing model. Each panel below the diagonal is the contour plot of the 2-D marginal posterior distribution for a given parameter pair. Panels on the diagonal are histograms of the 1-D marginal posterior distribution for each parameter (as per Fig. S4). **B.** Triangle plot of a typical variational solution returned by VBMC-VIQR.

### E.5 Sensitivity to imprecise noise estimates

In this paragraph, we look at how robust VBMC-VIQR is to different degrees of imprecision in the estimation of log-likelihood noise. We consider the same setup with three example problems as in the noise sensitivity analysis reported in main text (Fig. 4 in the main text). For this analysis, we fixed the emulated noise to $\sigma_{\text{obs}}(\boldsymbol{\theta}) = 2$ for all problems. We then assumed that the estimated noise $\widehat{\sigma}_{\text{obs}}(\boldsymbol{\theta})$, instead of being known (nearly) exactly, is drawn randomly as $\widehat{\sigma}_{\text{obs}} \sim \text{Lognormal}\left(\ln \sigma_{\text{obs}}, \sigma_\sigma^2\right)$, where $\sigma_\sigma \geq 0$ represents the jitter of the noise estimates on a logarithmic scale.

We tested the performance of VBMC-VIQR for different values of noise-of-estimating-noise, $\sigma_\sigma \geq 0$ (see Fig. S6). We found that up to $\sigma_\sigma \approx 0.4$ (that is, $\widehat{\sigma}_{\text{obs}}$ varying roughly between $0.5 - 2.2$ times the true value) the quality of the inference degrades only slightly. For example, at worst the MMTV metric rises from $0.13$ to $0.16$ on the Timing problem (less than $\sim 25\%$ increase), and in the other problems it is barely affected. These results show that VBMC-VIQR is quite robust to imprecise noise estimates. Combined with the fact that we expect estimates of the noise obtained from methods such as IBS to be very precise [27], imprecision in the noise estimates should not be an issue in practice.

Figure S6: **Sensitivity to imprecise noise estimates.** Performance metrics of VBMC-VIQR with respect to ground truth, as a function of noise-of-estimating-noise $\sigma_\sigma$. For all metrics, we plot the median and shaded areas are 95 % CI of the median across 50 runs. **A.** Absolute error of the log marginal likelihood (LML) estimate. **B.** Mean marginal total variation distance (MMTV).

### E.6 g-and-k model

We report here results of another synthetic test model omitted from the main text. The g-and-k model is a common benchmark simulation model represented by a flexible probability distribution defined

via its quantile function,

$$Q\left(\Phi^{-1}(p); \boldsymbol{\theta}\right) = a + b\left(1 + c\frac{1 - \exp\left(-g\Phi^{-1}(p)\right)}{1 + \exp\left(-g\Phi^{-1}(p)\right)}\right)\left[1 + \left(\Phi^{-1}(p)\right)^2\right]^k \Phi^{-1}(p), \qquad \text{(S22)}$$

where $a, b, c, g$ and $k$ are parameters and $p \in [0, 1]$ is a quantile. As in previous studies, we fix $c = 0.8$ and infer the parameters $\boldsymbol{\theta} = (a, b, g, k)$ using the synthetic likelihood (SL) approach [14, 28, 29]. We use the same dataset as [14, 29], generated with "true" parameter vector $\boldsymbol{\theta}_{\text{true}} = (3, 1, 2, 0.5)$, and for the log-SL estimation the same four summary statistics obtained by fitting a skew $t$-distribution to a set of samples generated from Eq. S22. We use $N_{\text{sim}} = 100$, which produces fairly precise observations, with $\sigma_{\text{obs}}(\boldsymbol{\theta}_{\text{MAP}}) \approx 0.14$. In terms of parameter bounds, we set LB $= (2.5, 0.5, 1.5, 0.3)$ and UB $= (3.5, 1.5, 2.5, 0.7)$ as in [14]; and PLB $= (2.6, 0.6, 1.6, 0.34)$ and PUB $= (3.4, 1.4, 2.4, 0.66)$.

Figure S7: **Performance on g-and-k model.** Performance metrics of various algorithms with respect to ground truth, as a function of number of likelihood evaluations, on the g-and-k model problem. For all metrics, we plot the median and shaded areas are 95% CI of the median across 100 runs. **A.** Absolute error of the log marginal likelihood (LML) estimate. **B.** Mean marginal total variation distance (MMTV). **C.** "Gaussianized" symmetrized Kullback-Leibler divergence (gsKL).

We show in Fig. S7 the performance of all methods introduced in the main text for three different inference metric: the log marginal likelihood (LML) loss, and both the mean marginal total variation distance (MMTV) and the "Gaussianized" symmetrized Kullback-Leibler divergence (gsKL) between approximate posterior and ground-truth posterior. For algorithms other than VBMC, we only report metrics they were designed for (posterior estimation for GP-IMIQR, model evidence for WSABI). The plots show that almost all algorithms (except WSABI) eventually converge to a very good performance across metrics, with only some differences in the speed of convergence. These results suggest that the g-and-k problem as used, e.g., in [14] might be a relatively easy test case for surrogate-based Bayesian inference; as opposed to the challenging real scenarios of our main benchmark, in which we find striking differences in performance between algorithms. Since we already present a simple synthetic scenario in the main text (the Ricker model), we did not include the g-and-k model as part of our main noisy-benchmark.

Finally, we note that when performing simulation-based inference based on summary statistics (such as here with the g-and-k model, and the Ricker model discussed in the main text), computing the marginal likelihood may not be a reliable approach for model comparison [30]. However, this is not a concern when performing simulation-based inference with methods that compute the log-likelihood with the entire data, such as IBS [27], as per all the other example problems in the main text.

## Supplementary references

[1] Acerbi, L. (2018) Variational Bayesian Monte Carlo. *Advances in Neural Information Processing Systems* **31**, 8222–8232.

[2] Jordan, M. I., Ghahramani, Z., Jaakkola, T. S., & Saul, L. K. (1999) An introduction to variational methods for graphical models. *Machine Learning* **37**, 183–233.

[3] Bishop, C. M. (2006) *Pattern Recognition and Machine Learning*. (Springer).

[4] Rasmussen, C. & Williams, C. K. I. (2006) *Gaussian Processes for Machine Learning*. (MIT Press).

[5] Acerbi, L. (2019) An exploration of acquisition and mean functions in Variational Bayesian Monte Carlo. *Proceedings of The 1st Symposium on Advances in Approximate Bayesian Inference (PMLR)* **96**, 1–10.

[6] O'Hagan, A. (1991) Bayes–Hermite quadrature. *Journal of Statistical Planning and Inference* **29**, 245–260.

[7] Ghahramani, Z. & Rasmussen, C. E. (2002) Bayesian Monte Carlo. *Advances in Neural Information Processing Systems* **15**, 505–512.

[8] Osborne, M., Duvenaud, D. K., Garnett, R., Rasmussen, C. E., Roberts, S. J., & Ghahramani, Z. (2012) Active learning of model evidence using Bayesian quadrature. *Advances in Neural Information Processing Systems* **25**, 46–54.

[9] Gunter, T., Osborne, M. A., Garnett, R., Hennig, P., & Roberts, S. J. (2014) Sampling for inference in probabilistic models with fast Bayesian quadrature. *Advances in Neural Information Processing Systems* **27**, 2789–2797.

[10] Kanagawa, M. & Hennig, P. (2019) Convergence guarantees for adaptive Bayesian quadrature methods. *Advances in Neural Information Processing Systems* **32**, 6234–6245.

[11] Knuth, D. E. (1992) Two notes on notation. *The American Mathematical Monthly* **99**, 403–422.

[12] Gessner, A., Gonzalez, J., & Mahsereci, M. (2019) Active multi-information source Bayesian quadrature. *Proceedings of the Thirty-Fifth Conference on Uncertainty in Artificial Intelligence (UAI 2019)* p. 245.

[13] MacKay, D. J. (2003) *Information theory, inference and learning algorithms*. (Cambridge University Press).

[14] Järvenpää, M., Gutmann, M. U., Vehtari, A., Marttinen, P., et al. (2020) Parallel Gaussian process surrogate Bayesian inference with noisy likelihood evaluations. *Bayesian Analysis*.

[15] Acerbi, L. & Ma, W. J. (2017) Practical Bayesian optimization for model fitting with Bayesian adaptive direct search. *Advances in Neural Information Processing Systems* **30**, 1834–1844.

[16] Krajbich, I., Armel, C., & Rangel, A. (2010) Visual fixations and the computation and comparison of value in simple choice. *Nature Neuroscience* **13**, 1292.

[17] Acerbi, L., Wolpert, D. M., & Vijayakumar, S. (2012) Internal representations of temporal statistics and feedback calibrate motor-sensory interval timing. *PLoS Computational Biology* **8**, e1002771.

[18] Acerbi, L., Dokka, K., Angelaki, D. E., & Ma, W. J. (2018) Bayesian comparison of explicit and implicit causal inference strategies in multisensory heading perception. *PLoS Computational Biology* **14**, e1006110.

[19] Goris, R. L., Simoncelli, E. P., & Movshon, J. A. (2015) Origin and function of tuning diversity in macaque visual cortex. *Neuron* **88**, 819–831.

[20] Akrami, A., Kopec, C. D., Diamond, M. E., & Brody, C. D. (2018) Posterior parietal cortex represents sensory history and mediates its effects on behaviour. *Nature* **554**, 368–372.

[21] Roy, N. A., Bak, J. H., Akrami, A., Brody, C., & Pillow, J. W. (2018) Efficient inference for time-varying behavior during learning. *Advances in Neural Information Processing Systems* **31**, 5695–5705.

[22] Carpenter, B., Gelman, A., Hoffman, M. D., Lee, D., Goodrich, B., Betancourt, M., Brubaker, M., Guo, J., Li, P., & Riddell, A. (2017) Stan: A probabilistic programming language. *Journal of Statistical Software* **76**.

[23] Haario, H., Laine, M., Mira, A., & Saksman, E. (2006) Dram: Efficient adaptive MCMC. *Statistics and Computing* **16**, 339–354.

[24] Neal, R. M. (2003) Slice sampling. *Annals of Statistics* **31**, 705–741.

[25] Gelman, A., Carlin, J. B., Stern, H. S., Dunson, D. B., Vehtari, A., & Rubin, D. B. (2013) *Bayesian Data Analysis (3rd edition)*. (CRC Press).

[26] Blei, D. M., Kucukelbir, A., & McAuliffe, J. D. (2017) Variational inference: A review for statisticians. *Journal of the American Statistical Association* **112**, 859–877.

[27] van Opheusden, B., Acerbi, L., & Ma, W. J. (2020) Unbiased and efficient log-likelihood estimation with inverse binomial sampling. *arXiv preprint arXiv:2001.03985*.

[28] Wood, S. N. (2010) Statistical inference for noisy nonlinear ecological dynamic systems. *Nature* **466**, 1102–1104.

[29] Price, L. F., Drovandi, C. C., Lee, A., & Nott, D. J. (2018) Bayesian synthetic likelihood. *Journal of Computational and Graphical Statistics* **27**, 1–11.

[30] Robert, C. P., Cornuet, J.-M., Marin, J.-M., & Pillai, N. S. (2011) Lack of confidence in approximate Bayesian computation model choice. *Proceedings of the National Academy of Sciences* **108**, 15112–15117.