[Reviews · NeurIPS 2020]

Review 1

Summary and Contributions: EDIT: thanks for the rebuttal. It has left my opinion of the paper unchanged. --- The authors present a framework for approximate Bayesian inference in models where the likelihood is expensive to evaluate and additionally corrupted by noise. The paper is an extension of Variational Bayesian Monte Carlo (Acerbi 2018) to the noisy setting. The method is essentially a two-step approach to approximate inference, carried out iteratively: 1. Build a surrogate for the log unnormalized posterior 2. Construct a variational approximation to the surrogate instead of the true posterior. The noisy setting requires more robust acquisition strategies for the active choice of locations in the input space where to evaluate the likelihood for constructing the surrogate. The introduction of such robust and effective acquisition functions is the key technical contribution of this paper.

Strengths: The authors are very clear about their contributions and the limitations of their work. The work addresses an important area, Bayesian inference for expensive and noise-corrupted likelihoods, and enhances previous work in this area. As such it is highly relevant to the NeurIPS community. A key achievement of the paper is a sound reassessment of acquisition functions for Bayesian quadrature to ensure a robust active learning scheme in the noisy setting. The claims are supported by a comparatively large set of real-world experiments.

Weaknesses: The authors assume the (heteroscedastic) noise to be given alongside the evaluations of the log unnormalized posterior. That could be restrictive for potential applications in which the noise level is unknown and a setup in which the noise is modelled could be desirable. One could further criticize the limitation of the method to relatively low input dimensions (max. ~10), but the authors do mention this issue as future direction of research.

Correctness: As far as I can tell, the statements made in the paper are correct, and the method provides plausible results, thus supporting the claims made.

Clarity: The paper is very well written and structured. It is, in fact, a pleasure to read! I particularly appreciate the details and useful background in the appendix, which provides all the material necessary to understand all relevant concepts used for VBMC. Thank you :)

Relation to Prior Work: Yes. The work directly builds on VBMC by Acerbi (2018) and extends this framework to noisy likelihoods. This requires the introduction of new/modified acquisition functions for an effective active learning scheme for the Bayesian quadrature part.

Reproducibility: Yes

Additional Feedback: I really appreciated the well-structured paper and the supporting material, which supports understanding rather than just being dumped additional material. **Questions:** - In synthetic likelihood experiments, you say that the noise level is estimated from repeated evaluations. But these should in practice be expensive to obtain, right? - Could you comment on the assumption that the noise level is known? What additional challenges arise when inferring observation noise (which you mention as an option, but don't elaborate further on)? **Comments:** - l. 159 $\Phi$ refers to the cumulative Gaussian? I don't think you say that anywhere.


Review 2

Summary and Contributions: This paper extends Variational Bayesian Monte Carlo method to the case of noisy log-likelihood evaluations, using new 'global' acquisition functions to guarantee robustness against noise. Experiments on real datasets from neuroscience suggests that the new method outperforms existing variants of inference methods using 'local' acquisition functions. ------- Update: I have decided to stick to my decision.

Strengths: 1. The paper is well-written, with contributions and the status of prior work clearly stated. It also makes use of rigorous mathematical formulations to describe the problem to be addressed. 2. The experiments are comprehensive, and largely align with the claims in the method description section.

Weaknesses: 1. There seems to be a lack of theoretical guarantee other than empirical evaluation. Would it be possible to obtain ELBO-like bounds for both choices of global acquisition functions? 2. The methods suggested seem to be built on existing work(e.g. [11], [10]). The motivations for modifying IMIQR equation (7) to equation (8) is not directly obvious, and other than the variational formulation of equation (7), it is not immediately clear whether there are other technical novelties. It would be greatly appreciate if the authors could elaborate on the novelty of the methods more. 3. Small grammatical errors exist(e.g. suggest to sample points that maximize EIG ... in section 3.2).

Correctness: Yes. The methodology involved is valid.

Clarity: Yes. There is no noticeable problem in style and structure.

Relation to Prior Work: Yes.

Reproducibility: Yes

Additional Feedback:


Review 3

Summary and Contributions: This paper considers the problem of Bayesian inference with black-box non-cheap likelihoods. The paper extends an earlier approach, VBMC, to the scenario where the likelihood evaluations are noisy. Earlier VBMC in [5,6] assumed exact likelihood evaluations and the main idea of it is to model the log-likelihood with Gaussian process regression which is further combined with variational inference and Bayesian quadrature. New likelihood evaluations are selected sample-efficient way using active learning techniques. The resulting algorithm produces a Gaussian mixture approximation to the posterior and an estimate for the marginal likelihood. What is new in this paper are new acquisition functions designed for the noisy evaluations, a variational whitening approach that can apparently improve posterior approximation and an extensive benchmark with models from computational and cognitive neuroscience, ranging from 3 to 9 parameters, and applied to real datasets. The Supplementary contains further details on the algorithm, acquisition functions etc. and further experimental results. Update after rebuttal: ################# Thanks for the rebuttal. I mainly agree on the answers regarding the first three points (i.e. the noise, scaling to high dimension and theoretical guarantees). The motivation behind VIQR makes more sense to me now but the justification for it nevertheless feels very heuristic. I hope the authors can further improve their explanation and state truthfully in the paper that this is the case. I am however not very happy with the answer I got regarding the comparison to other methods; I would have expected to see more analysis on this. It would be important to know why certain methods don't work so that future work can perhaps improve on them. My overall opinion about the paper remains the same.

Strengths: Extending the VBMC algorithm to handle noisy likelihood evaluations is a well-motivated problem. Namely, when the likelihood evaluations are expensive and possibly noisy, MCMC and standard variational methods become infeasible so that more sophisticated methods are needed yet not many algorithms have been proposed for this setting. The extended version of the VBMC software would be useful for practitioners. The algorithm seems quite complex though, with various heuristics and tricks involved but I realise they might be needed to ensure robustness when dealing with complex real-world models. Some of the heuristics are further empirically investigated in the Supplementary. The experiments are extensive: 7 models are considered and two state-of-the-art baseline are included. The experiments show that the proposed method outperforms the baselines and works consistently well in all the real-world models considered.

Weaknesses: From a purely technical viewpoint, the proposed new features (variational whitening, acquisition functions for the noisy case) are not very exciting or follow from the related research in a fairly straightforward way. The motivation/justification of the acq functions is not well explained. The method in section 3.1 seems heuristic, 3.2 makes sense, the justification for variational interquantile range (VIQR) in 3.3. is unclear. It seems that the Bayesian decision theoretic justification given in [10] for their GP-IMIQR method does not follow to the VBMC case and it is consequently questionable why the resulting method is still related to the IQR and called VIQR. VIQR seems to work well in practice though so I would suggest to improve the text here with a more clear justification especially for the VIQR method (and consider renaming it, if appropriate and feasible). VBMC-IMIQR and VBMC-VIQR clearly work better than WSABI and GP-IMIQR methods as shown in fig 2 and 3. However, it remains quite unclear where the improvements come from and whether the comparison is fair. E.g. in the fig3 scenario I wonder if it is the Variational inference approach, possibly more careful implementation with various heuristics (which are apparently not present in the competing method) or a more suitable GP model, that yields the improved performance as compared to gp-imiqr. It is a bit puzzling that IMIQR acq function always works well in the case of VBMC yet GP-IMIQR works poorly on some problems - especially in the neuronal model - and fairly well in others. I would have hoped to see more analysis on this; e.g. it should be easy to check what kind of evaluation locations are obtained with various methods to get some idea why the competing method is failing. Similarly, the reason why WSABI works very poorly on all problems could have been better investigated/explained as I find this result somewhat surprising. As in the original VBMC, the fact that the joint density is both modeled with GP and also approximated with Gaussian mixture is awkward.

Correctness: The claims and the methods seem correct, with the exception of the dilemma with VIQR explained above. The experiments seem carefully done.

Clarity: Mostly yes. However, some of the discussion (especially on the acquisition functions in Section 3) could be in a much clearer way. There are many details and heuristics involved and reading through them would be very tedious especially for someone not already familiar with original VBMC (but this seems unavoidable).

Relation to Prior Work: Yes. "Variational Bayes with Synthetic Likelihood" https://arxiv.org/pdf/1608.03069.pdf seems relevant and could be cited though.

Reproducibility: Yes

Additional Feedback: 60: batches are mentioned here but the acquisition functions introduced later on seem to focus on non-batch case 98-100: A bit unclear sentence 136: should this read: evaluate where the pointwise variance of ... is maximum? 137: Might be good to clarify this sentence as the claim holds only in the non-noisy case, if I have understood correctly 158: [10] derives their acquisition function from Bayesian decision theory so saying that it is *defined* this way seems inaccurate. The prior density is missing from (7) as compared to (30) in [10] and I wonder why 272: Computing marginal likelihood for simulation models may not be a reliable approach for model comparison, please see "Lack of confidence in ABC model choice" https://arxiv.org/abs/1102.4432 Might be good to briefly mention this in the case of Ricker model. I wonder how VBMC would handle multimodal posteriors with separated modes or highly banana-shaped posteriors. Do the real-world models considered feature such posteriors? Perhaps gp-imiqr might work better for such cases? I noticed that some properties of the test problems are shown Figure S4 and S5 but it would be good to describe the shapes of the posteriors in the main text. I wonder if the proposed acquisition functions are also useful for the non-noisy situation of the original VBMC paper.


Review 4

Summary and Contributions: In this paper, the approximate Bayesian inference problem for noisy, black-box, non-cheap likelihoods. Several acquisition functions and variational whiting trick is proposed and tested on a number of models and real datasets from computational neuroscience. ####### Thanks for the rebuttal. After reading the response and also the comments from other reviewers, I decided to raise my score from to 6.

Strengths: The paper is clearly written ,well organized and very easy to understand. The intuition behind the proposed acquisition functions are well explained. The experiments are thorough and the results are significant to support the claims of the paper.

Weaknesses: However, the proposed method still has several weaknesses that concern me. 1, The proposed method lacks theoretical grounding. How does the surrogate ELBO relate to the true model likelihood? Given the proposed acquisition functions, is the model ELBO guaranteed to be improved? How about the convergence property? 2, Lack of ablation study. Several tricks seem to be crucial to the result. For example, the impact of the whitening trick is not studied. How does this trick contribute to the improvement? Without relevant analysis, I am not sure if it is justified to claim that the improvements are mostly due to the new acquisition functions. Also, the method assumes the knowledge of (at least an estimate of \sigma_obs). How does the estimation error affect the results? Will it change the claims of the paper? 3, I am not fully sure the applicability of the proposed method. The scalability of the proposed method is unclear. I guess for high dimensional parameter space, it is unfeasible to use GP models to approximate the likelihood functions, right? For example, imagine that the model/simulator has a Bayesian neural network as one of the components. How does the method perform in this case?

Correctness: The methodology looks correct to me. Although it still lacks relevant theoretical analysis.

Clarity: Yes.

Relation to Prior Work: Yes.

Reproducibility: Yes

Additional Feedback:

[Author Response · NeurIPS 2020]

We thank the reviewers for their useful and thoughtful feedback. We are glad to see that our work was found "*highly*
*relevant to the NeurIPS community*" (**R1**) and to be addressing "*a well-motivated problem*" (**R3**) in "*an important area*"
(**R1**); that our empirical validation was "*sound*" (**R1**), "*comprehensive*" (**R2**), "*extensive: 7 models are considered and*
*two state-of-the-art baselines are included*" (**R3**), and "*thorough (**R4**)*"; with our claims "*supported by a comparatively*
*large set of real-world experiments*" (**R1**), results that "*largely align with the claims*" (**R2**) and "*are significant to support*
*the claims of the paper*" (**R4**). That is, "*the experiments show that the proposed method outperforms the baselines and*
*works consistently well in all the real-world models considered*" (**R3**). We address the reviewers' comments below and
incorporated all feedback in the revised paper. Here with 'VBMC' we refer to our method for noisy inference.

**Known noise (R1, R4).**    The assumption that log-likelihood noise $\sigma_{\mathrm{obs}}$ is (approximately) known is less limiting than it
might seem. The common *synthetic likelihood* approach already needs multiple samples to build the multivariate normal
pdf used to estimate the likelihood, hence $\sigma_{\mathrm{obs}}$ can be easily estimated via bootstrap with negligible cost (as shown in
[10]). *Inverse binomial sampling* (IBS, [41]), used extensively in our analyses, automatically provides both an estimator
of the log-likelihood and of $\sigma_{\mathrm{obs}}$ for free. For other techniques, bootstrap is often a feasible and easy-to-implement
choice. **R4:** "*How does the estimation error* [of $\sigma_{\mathrm{obs}}$] *affect the results? Will it change the claims of the paper?*" Great
question. We performed a new analysis by rerunning VBMC on several problems (aDDM, Timing, Neuronal) while
drawing the estimated $\widehat{\sigma}_{\mathrm{obs}} \sim \mathrm{Lognormal}\left(\ln \sigma_{\mathrm{obs}}, \sigma_{\sigma}^2\right)$ for increasing values of noise-of-estimating-noise, $\sigma_{\sigma} \geq 0$. At
worst, the performance of VBMC via the MMTV metric degrades only by $\sim 0.03$ points on average (e.g., from $0.13$ to
$0.16$ on the Timing problem; see Fig. 3 for reference), with $\sigma_{\sigma}$ up to $0.4$ (i.e., $\widehat{\sigma}_{\mathrm{obs}}$ roughly between $0.5 - 2.2$ times the
true value); showing that VBMC is very robust to imprecise estimates of the noise. Thus, our claims are unaffected.

**Dimensions & Applicability (R1, R4).**    The limit of VBMC to $D \sim 10$ input dimensions is common to approaches
that use GP surrogates (e.g., Bayesian optimization [22]), and an open area of research which we intend to pursue. Still,
*plenty* of models in computational biology and neuroscience have up to $\sim 10$ parameters, so there is *wide applicability*
of our method. As proof, consider the widespread usage within computational neuroscience and related fields of the
recent BADS toolbox [26] for (noisy) Bayesian optimization, which shares similar limitations as VBMC in terms of $D$.
Crucially, VBMC is a large step forward with respect to BADS for practitioners in the field in that it affords *full Bayesian*
*inference* (as opposed to limited to point estimation), so we expect it to impact a wide audience. Finally, while our
method does not directly apply to high-$D$ machine learning models (e.g., Bayesian neural networks), it could be used to
infer posteriors over *hyperparameters* of ML models, which we see as a relevant research direction.

**Theoretical guarantees (R2, R4).**    While we share the reviewers' desire for convergence guarantees, we also note that
convergence proofs of adaptive Bayesian quadrature methods are outstanding theoretical contributions in themselves.
For example, despite a growing body of work on adaptive methods over the years, *only last year* Kanagawa and Hennig
[37] were first able to prove convergence for a class of *local* acquisition functions in Bayesian quadrature. By contrast,
our paper adds to the literature of solid empirical contributions with theoretically-motivated choices. We believe that our
strong empirical validation, judged very positively by all reviewers, while not at all replacing a mathematical derivation,
should provide confidence in our method and inspire future theoretical research on VBMC.

**Motivation of $a_{\mathbf{VIQR}}$ acquisition function (R2, R3).**    The rationale for going from $a_{\mathrm{IMIQR}}$ (Eq. 7) to $a_{\mathrm{VIQR}}$ (Eq. 8) is:
(1) sampling from the posterior $\exp(\overline{f}(\boldsymbol{\theta}))$ is relatively hard, whereas sampling from $q_\phi(\boldsymbol{\theta})$ (the variational posterior) is
trivial; (2) $q_\phi(\boldsymbol{\theta})$, by construction, approximates $\exp(\overline{f}(\boldsymbol{\theta}))$ up to a normalizing constant (irrelevant for optimization).
Thus, we go from Eq. 7 to Eq. 8 by swapping $\exp(\overline{f}(\boldsymbol{\theta}))$ with $q_\phi(\boldsymbol{\theta})$, with substantial gains (see Table 1). While $a_{\mathrm{VIQR}}$
adds a layer of variational approximation ($q_\phi(\boldsymbol{\theta}) \approx \exp(\overline{f}(\boldsymbol{\theta}))$), hence the name, it still directly approximates $a_{\mathrm{IMIQR}}$.

**Response to remaining comments.**    **R1:** *Modelling* $\sigma_{\mathrm{obs}}$, while possible, is tricky in practice due to the trade-off
between $\sigma_{\mathrm{obs}}$ and variability of the latent function, which may be hard to disambiguate. However, as argued above,
obtaining (approximate) *estimates* of $\sigma_{\mathrm{obs}}$ is very often a viable solution; we leave modeling of *unknown* $\sigma_{\mathrm{obs}}$ as future
work. **R2:** In terms of novelty, we propose *variational whitening* (which improves performance on hard problems); we
extend (in a fairly straightforward way) previous work from [10,11] to the VBMC framework; we build a novel, extensive
noisy benchmark with many models and real datasets from computational and cognitive neuroscience, showing that
VBMC vastly outperforms state-of-the-art, providing a meaningful contribution to the field. **R3:** We discussed potential
reasons of failure for WSABI and GP-IMIQR in the Supplement, D.2. We believe we performed a fair comparison, in that
for both WSABI and GP-IMIQR we tried to fix potential issues, and report the best-performing variants. Re. multimodal
posteriors, VBMC (similarly to MCMC) would have trouble with far, disconnected modes, and it is an active research
area; the exploratory tendency of GP-IMIQR might help here, but also lead to instabilities. Re. the noiseless case, see Fig.
S6, in which we varied the amount of noise; interestingly, most acquisition functions perform similarly with no noise
($\sigma_{\mathrm{obs}} \approx 0$). In Eq. 7, the prior density is included in $\exp(\overline{f})$ ($f$ models the log-joint). **R4:** "*Lack of ablation study.* [...]"
The ablation studies are mentioned in the text, and fully reported in the Supplement (E.3, 'Lesion study'). In particular,
removing variational whitening degrades performance on the difficult Neuronal and Rodent problems. However, we
found no major differences in performance on the other problems; leading us to claim that the improved performance of
VBMC on noisy problems (wrt. the original framework [5]) is mostly due to the new acquisition functions.

[Meta-Review · NeurIPS 2020]

This work extends an existing inference algorithm to the case of noisy or expensive likelihood evaluations. Reviewers expressed concern that the work is a bit incremental, but is thorough and addresses an important problem setting. Additionally, the paper could be improved by better more clearly motivating the techniques used in the proposed method.